# Understanding the Robustness of Distributed Self-Supervised Learning Frameworks Against Non-IID Data

**Xuanyu Chen**[1]**, Nan Yang**[1*]**, Shuai Wang**[2]**, Dong Yuan**[1*]
[1]School of Electrical and Computer Engineering, The University of Sydney
[2]Northwestern Polytechnical University
{xuanyu.chen,n.yang,dong.yuan}@sydney.edu.au
{s.wang}@mail.nwpu.edu.cn

## Abstract

Recent research has introduced distributed self-supervised learning (D-SSL) approaches to leverage vast amounts of unlabeled decentralized data. However, D-SSL faces the critical challenge of data heterogeneity, and there is limited theoretical understanding of how different D-SSL frameworks respond to this challenge. To fill this gap, we present a rigorous theoretical analysis of the robustness of D-SSL frameworks under non-IID (non-independent and identically distributed) settings. Our results show that pre-training with Masked Image Modeling (MIM) is inherently more robust to heterogeneous data than Contrastive Learning (CL), and that the robustness of decentralized SSL increases with average network connectivity, implying that federated learning (FL) is no less robust than decentralized learning (DecL). These findings provide a solid theoretical foundation for guiding the design of future D-SSL algorithms. To further illustrate the practical implications of our theory, we introduce MAR loss, a refinement of the MIM objective with local-to-global alignment regularization. Extensive experiments across model architectures and distributed settings validate our theoretical insights, and additionally confirm the effectiveness of MAR loss as an application of our analysis.

## 1 Introduction

Deep learning advancements have been driven by large-scale datasets, as seen in the training of LLMs, which require billions of data points (Hoffmann et al., 2022; Rae et al., 2021). However, real-world data is often decentralized, such as surveillance footage from distributed security cameras. This abundance of unlabeled distributed data has spurred interest in distributed self-supervised learning (D-SSL) (Zhuang et al., 2021a; Wang et al., 2022), which extends self-supervised learning (SSL) to decentralized settings. Existing D-SSL frameworks can generally be distinguished in two aspects: differing by the adopted self-supervised learning (SSL) method or by the applied distributed framework. Self-supervised learning (SSL) is a widely used technique to learn representations without human-labeled annotations by solving pretext tasks that generate supervisory signals from raw data (Gui et al., 2024). Depending on the approach used to generate supervisory signals, SSL methods are broadly categorized into Contrastive Learning (CL) and Masked Image Modeling (MIM) (Liu et al., 2021; Zhang et al., 2022), with representative methods like SimSiam (Chen & He, 2021) and MAE (He et al., 2022). On the other hand, federated learning (FL) and decentralized learning (DecL) are two main frameworks in training models with distributed data (Verbraeken et al., 2020; Sun et al., 2024). FL aggregates local models via a central server (McMahan et al., 2017a; Zhuang et al., 2021a), while DecL enables direct inter-client communications for aggregating models, enhancing privacy and avoiding the dependence on the central server (Tang et al., 2022; Ayache & El Rouayheb, 2019).

One unique challenge of D-SSL research is handling highly heterogeneous data on clients. Distributed data among multiple clients are normally non-independent and identically distributed (non-IID), leading to performance degradation (Zhu et al., 2021). To tackle this challenge, previous works

---

*Corresponding Authors: Nan Yang and Dong Yuan.

proposed advanced D-SSL algorithms with robustness to heterogeneous data. Notable examples include FedU (Zhuang et al., 2021a), Orchestra (Lubana et al., 2022), and L-DAWA (Rehman et al., 2023). However, despite continuous algorithmic innovation, there is still a lack of theoretical understanding of this heterogeneity problem. For example, FedU was designed within the FL framework, but how would its robustness to non-IID data change if deployed in a DecL framework without coordination from the server? Similarly, state-of-the-art D-SSL algorithms are primarily based on CL, while the adaptation of MIM methods to distributed settings remains under-explored. Could D-SSL based on MIM offer greater robustness to non-IID data than CL-based methods? These confusions converge into a fundamental research question affecting the advancement of D-SSL:

*How robust are different D-SSL frameworks against data heterogeneity?*

To address this question, this paper aims to provide a theoretical understanding of how different D-SSL frameworks behave under heterogeneous data. We construct mathematical models in a simplified non-IID setting and rigorously analyze the representability of local and global representations learned by these algorithms. Our analysis reveals two key insights: (i) D-SSL algorithms based on Masked Image Modeling (MIM) are inherently more robust than those based on Contrastive Learning (CL), although their robustness still degrades under severe divergence between local and global distributions; and (ii) the robustness of decentralized SSL improves with the average connectivity of the network, which suggests that decentralized SSL is only as robust as federated D-SSL in the limited case of full connectivity (i.e., a fully connected network). Building on these insights, we also explore how theoretical results can inform algorithmic design. As an illustration, we refine the MIM objective with the additional alignment regularization, which we call MAR loss, to encourage local-to-global representation consistency. Finally, we conduct extensive experiments on ResNet (He et al., 2016) and Vision Transformer (ViT) (Dosovitskiy et al., 2020) across a variety of distributed settings and benchmark datasets to validate our theoretical findings and to demonstrate the usefulness of MAR loss as a practical example.

In summary, our main contributions are listed below:

1. We develop a rigorous theoretical analysis of distributed self-supervised learning (D-SSL) under non-IID data, showing that MIM-based D-SSL is inherently more robust than CL-based D-SSL.

2. We establish the relationship between network connectivity and robustness, proving that decentralized SSL benefits from higher connectivity and that federated SSL is no less robust than decentralized SSL.

3. We introduce MAR loss as an illustrative case study demonstrating how our theoretical results can guide algorithmic design, by refining the MIM objective with alignment regularization.

4. We conduct extensive experiments across model architectures and distributed settings, which validate our theoretical insights and further confirm the effectiveness of MAR loss.

## 2 RELATED WORK

**Self-Supervised Learning.** Self-supervised learning (SSL) leverages unlabeled data by generating pseudo labels from raw inputs to learn meaningful representations (Gui et al., 2024). Vision-based SSL methods are typically categorized into contrastive learning (CL) and masked image modeling (MIM) (Zhang et al., 2022; Liu et al., 2021). CL learns representations by maximizing the similarity between positive pairs (i.e., similar data points created by data augmentation) and minimizing it between negative pairs (i.e., data pairs created by other data points) (Chen et al., 2020; He et al., 2020). Recent methods like SimSiam (Chen & He, 2021) and BYOL (Grill et al., 2020) advance the original contrastive loss by removing terms related to negative pairs, which improves stability and reduces batch size dependence. MIM, in contrast, randomly masks out patches of input images and predicts the missing parts, learning representations through a reconstruction loss (Bao et al., 2021; Zhou et al., 2021; Xie et al., 2022; He et al., 2022). Although different in formulation, recent studies have shown that many MIM methods have close connections to CL (i.e., their objectives can be directly re-formulated as contrastive loss (Zhang et al., 2022; Kong et al., 2019)). In this work, we aim to figure out which SSL paradigm is inherently more robust against data heterogeneity.

**Distributed Learning.** Distributed learning enables collaborative model training across multiple clients without sharing data. Two dominant frameworks in this area are: federated learning (FL), which uses a central server to coordinate and aggregate models (McMahan et al., 2017a), and decentralized learning (DecL), where clients exchange models locally with neighbors (Tang et al., 2022; Ayache & El Rouayheb, 2019). While FL is more widely adopted (Zhang et al., 2021) for better convergence and training effectiveness, DecL offers benefits in scalability and privacy. Recent studies have started comparing these two frameworks (Beltrán et al., 2023; Hegedűs et al., 2021). For example, Sun et al. explored which leads to better generalization and the impact of network architecture on generalization (Sun et al., 2024). However, the relationship between network architecture and the non-IID robustness in distributed settings is still unclear. Our work addresses this gap by providing both theoretical analysis and empirical findings to clarify this relationship.

**Distributed SSL.** Distributed SSL (D-SSL) integrates SSL with distributed frameworks to leverage unlabeled, decentralized data while preserving privacy (Zhuang et al., 2021a; Yang et al., 2023). A core challenge is learning robust representations under data heterogeneity (Zhu et al., 2021). Prior work has primarily focused on algorithmic solutions such as FedU (Zhuang et al., 2021a) and L-DAWA (Rehman et al., 2023). Although some studies also provide theoretical analyses, their purpose is to demonstrate the validity of the proposed algorithms rather than to advance the understanding of the robustness variance between different D-SSL frameworks (Lubana et al., 2022; Jing et al., 2024). The most relevant theoretical work is by Wang et al., who showed that SSL is more robust than supervised learning in distributed settings (Wang et al., 2022). Unfortunately, their study only analyzed a specific case of D-SSL where CL is combined with FL and did not extend it to other types of D-SSL frameworks. In contrast, our work delves deeper into these differences, shedding light on understanding the insensitivity of various D-SSL approaches under heterogeneous conditions.

## 3 PROBLEM SETUP

To provide theoretical insights on understanding this central question, we first introduce our problem setup about distributed training and D-SSL with heterogeneous data.

### 3.1 DISTRIBUTED TRAINING

**Distributed Setting.** Consider a distributed scenario consisting of a connected network of N clients, represented as a graph $\mathcal{G} = (\mathcal{V}, \mathcal{E})$, where $\mathcal{V}$ is the set of clients and $\mathcal{E}$ is the set of edges denoting direct communication links between clients. The connectivity of the graph is captured by a matrix $A \in \mathbb{R}^{N \times N}$, referred to as the adjacency matrix, where $A_i$ denotes the set including client $i \in [N]$ itself and its neighbors shown by $\mathcal{E}$, $|A_i|$ represents the size of this neighborhood set or the connectivity of client $i$, and $|\bar{A}| = \frac{1}{n} \sum_{i=1}^{n} (|A_i|)$ is the average connectivity. Hence, distributed training conducted through DecL satisfies $\forall i \in [N], 2 \leq |A_i| \leq N$. In contrast, FL relies on a central server that aggregates local models from all clients and broadcasts the global model back to them in each round, as in FedAvg (McMahan et al., 2017a). This architecture effectively enables every client to communicate with all others through the server, which corresponds to a fully connected decentralized topology where $\forall i \in [N], |A_i| = N$. A more formal specification of the graph structure and the mixing-weight conditions for this distributed setting is provided in Appendix A.6.1.

**Objective of Distributed Optimization.** To utilize different clients to learn useful representations, distributed training generally optimizes the below global objectives:

$$W_{Dec}^* = \min_W \frac{1}{N} \sum_{i=1}^{N} \frac{1}{|A_i|} \sum_{j \in A_i} \mathcal{L}_j(W_j); \quad W_{Fed}^* = \min_W \frac{1}{N} \sum_{i=1}^{N} \mathcal{L}_i(W_i) \tag{1}$$

where $\mathcal{L}_j$ is the objective of local SSL on client $j$, $W_{Dec}^*$ and $W_{Fed}^*$ denote the global objective of DecL and FL, respectively. In particular, at each iteration of DecL, each client conducts local updates using the local dataset and aggregates the updated local model with those from neighbors (Tang et al., 2022). For generating the global model for downstream tasks, there will be an additional aggregation on all local models after all iterations. Differently, the optimization of FL involves each round of model aggregation only on the central server (McMahan et al., 2017a). Then, the server broadcasts the global model to all clients for the next round of training. Note that the FL framework does not need another aggregation between all local models since the updated global model on the server can be used directly for fine-tuning.

## 3.2 RIGOROUS ANALYSIS OF D-SSL ON A SIMPLIFIED NON-IID SETTING

**Non-IID Client Data.** D-SSL involves all clients collaboratively training a global model by leveraging their local unlabeled datasets $\{D_i\}_{i=1}^N$ and communicating over the graph $\mathcal{G}$. Since sharing data is prohibited to protect privacy, the heterogeneity across these distributed data sources generally leads to a performance drop in many distributed applications (Zhuang et al., 2021a; McMahan et al., 2017a). Two common types of data heterogeneity are: feature heterogeneity and label heterogeneity (Zhu et al., 2021). In this paper, we follow previous works (Wang et al., 2022; Liu et al., 2022) to model a simplified but formal label non-IIDness between local datasets as follows.

The global data distribution $D = \bigcup_{i=1}^N D_i$ across clients is assumed to contain unlabeled data from $2N$ classes. For the dataset on client $i$, the local data distribution $D_i$ is constrained and imbalanced on three classes, with most samples belonging to classes $2i-1$ and $2i$, while the remaining very few samples come from the class $h_i \in [2N] \setminus \{2i-1, 2i\}$. Specifically, for a sufficiently large positive integer $d > 0$, let $x \in \mathbb{R}^d \sim D_i$ be the data points in the local

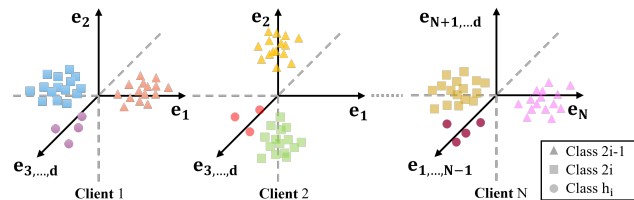

Figure 1: Illustration of the constructed heterogeneous distribution for local data on clients. Each client holds two unique data classes.

dataset and $e_1, \ldots, e_d$ be the standard unit-norm vectors of the d-dimensional Euclidean space. For class $2i-1$, we set $x^{(2i-1)} = e_i - \Sigma_{k \neq i, k=1}^N q^{(2i-1,k)} \tau e_k + \mu \xi^{(2i-1)}$, where $\tau$ and $\mu$ are two positive hyperparameters, $q$ is sampled uniformly from $\{0, 1\}$ and $\xi \sim \mathcal{N}(0, I)$ from Gaussian distribution. Likewise, for class $2i$, we define $x^{(2i)} = -e_i - \Sigma_{k \neq i, k=1}^N q^{(2i,k)} \tau e_k + \mu \xi^{(2i)}$. The size of the data from classes $2i-1$ and $2i$ are equal and both grow in polynomials of $d$. For infrequent class $h_i$, the samples are generated as: $x^{(h_i)} = e_{h_i} + \mu \xi^{(h_i)}$ and the amount of data is sublinear in $d$, denoted as $O(d^\alpha)$ with $\alpha \in (0, 1)$). Furthermore, we assume all $N$ local datasets to have an equal total number of samples, i.e., $|D_1| = |D_2| = \ldots = |D_N|$. To facilitate understanding, we provide an overview of this non-IID data distribution in Figure 1. Next, we consider CL and MIM as two main paradigms of SSL and formulate CL and MIM, respectively.

**CL Formulation.** For CL, we adopt the more advanced Simsiam (Chen & He, 2021) which trains with only the positive pairs $(g_a(x), g_b(x))$, where $g_a(\cdot)$ and $g_b(\cdot)$ are random augmentations drawn from SimSiam's augmentation policy (e.g., Gaussian noise, flipping). Consider a linear embedding function $f_W(x) = Wx$, where the weight matrix $W$ satisfies $W \in \mathbb{R}^{c \times d}$ and $c \geq 2N$ according to the distributed settings, the local objective on client $i$ is defined as:

$$\mathcal{L}_{CL} = -\mathbb{E}_{x \sim D_i}[|(W(g_a(x)))^\intercal(W(g_b(x)))] + \frac{1}{2}||W^\intercal W||_F^2. \tag{2}$$

Eq.(2) captures the SimSiam loss by utilizing the negative inner product $\langle a, b \rangle$ to measure the distance between the positive pairs. This objective also excludes a feature predictor for simplicity and includes a regularization term $||W^\intercal W||_F^2$ for more mathematically tractable, similar to previous works (Wang et al., 2022; Liu et al., 2022). Note that Eq.(2) stands for a general form of Simsiam loss due to the wide class of augmentation functions (Gui et al., 2024). For a detailed and tractable theoretical exploration, we consider the linear formulation of data augmentation and further differ CL by the similarity between $g_a(\cdot)$ and $g_b(\cdot)$. In particular, for the case where the positive pairs are generated by similar augmentations, the objective becomes $\mathcal{L}_{CL} = -\mathbb{E}_{x \sim D_i}[(W(x + \xi))^\intercal(W(x + \xi'))] + \frac{1}{2}||W^\intercal W||_F^2$, where $\xi, \xi' \sim \mathcal{N}(0, I)$ are random noise sampled IID from the Gaussian distribution. On the other hand, when $g_a(\cdot)$ and $g_b(\cdot)$ are different, we define the loss as $\mathcal{L}'_{CL} = -\mathbb{E}_{x \sim D_i}[(W(x + \xi))^\intercal(W(Hx))] + \frac{1}{2}||W^\intercal W||_F^2$, where $H \in \mathbb{R}^{d \times d}$ denotes a linear image transformation (e.g., rotation, translation, etc.). The formal conditions on $H$ are given in Appendix A.6.1.

**MIM Formulation.** For MIM, a random binary mask $m \in \{0, 1\}^d$ (created by uniformly sampling 0 with probability $p$, i.e., mask ratio) is applied to partition the input $x$ into two complementary views: the unmasked part $x_1 = x \odot m$ and the masked part $x_2 = x \odot (1 - m)$ satisfying $x_1 + x_2 = x$. Then, we train an encoder-decoder model $f = f_d \circ f_e$, where the encoder $f_e$ encodes the input $x_1$ to a latent representation $z = f_e(x_1)$, and the decoder $f_d$ decodes $z$ back to pixel space to reconstruct the

masked part $x_2$. Hence, considering a linear encoder and decoder with embedding matrix $W_e \in \mathbb{R}^{c \times d}$ and $W_d \in \mathbb{R}^{d \times c}$, the local objective of MIM is given by

$$\mathcal{L}_{MIM} = \mathbb{E}_{x \sim D_i} \mathbb{E}_{x_1, x_2 | x} ||f_d(f_e(x_1)) - x_2||^2 = \mathbb{E}_{x \sim D_i} ||W_d W_e (x \odot m) - (x \odot (1-m))||^2, \quad (3)$$

where the mean square error (MSE) loss is utilized to enforce the reconstructed image to be similar to the original image, and $\odot$ denotes the Hadamard product. Recent studies have focused on the connection between MIM and contrastive losses and found that the MIM reconstruction objective admits an alignment between the masked and unmasked parts (Zhang et al., 2022; Kong et al., 2019). Based on these results, we adopt an alignment-style formulation of Eq.(3) with $W := W_e \in \mathbb{R}^{c \times d}$:

$$\mathcal{L}_{MIM} = -\mathbb{E}_{x \sim D_i}[(W(x \odot m))^\intercal (W(x \odot (1-m)))] + \frac{1}{2}||W^\intercal W||_F^2, \quad (4)$$

which implicitly aligns the masked and unmasked views in the embedding space. The regularization term $||W^\intercal W||_F^2$ is also introduced to ensure a well-posed quadratic form and improve the traceability.

## 4 THEORETICAL INSIGHTS

In this section, we use the above problem setup to model different D-SSL frameworks and compare their robustness to heterogeneous data. Differences in applied SSL and network architecture lead to distinctions in learned representations, which can be further explored to determine variance in robustness. Due to page limitations, the complete proof for our analysis is provided in Appendix A.6.

### 4.1 ANALYSIS OF REPRESENTATIONS LEARNED BY D-SSL

We begin our analysis with the definition of the representability of the learned representation.

**Definition 4.1.** (Representability Vector (RV)). Let $\{e_1, \ldots, e_d\}$ be the standard basis of $\mathbb{R}^d$. Let $W = [w_1, \ldots, w_c]^\intercal \in \mathbb{R}^{c \times d}$ be the feature matrix learned by the linear embedding function $f_W(x) = Wx$, where $c \leq d$. For row space $\mathcal{R} = \text{row}(W) \subseteq \mathbb{R}^d$, we denote the representability of $\mathcal{R}$ as a vector $r = [||\Pi_\mathcal{R}(e_1)||_2^2, \ldots, ||\Pi_\mathcal{R}(e_d)||_2^2]^\intercal$, where $\Pi_\mathcal{R}(e_k)$ is the projection of $e_k$ onto $\mathcal{R}$ for $k \in [d]$. Hence, we have $||\Pi_\mathcal{R}(e_k)||_2^2 = \sum_{j=1}^c (e_k^\intercal v_j)^2$, where $\{v_1, \ldots, v_c\}$ is any orthonormal basis of $\mathcal{R}$.

The intuition behind this definition is that for any input vectors $x \in \mathbb{R}^d$, the learned feature space should have a good representation of the standard basis vectors, $e_1, \ldots, e_d$, to perform well. In particular, these basis vectors should have large projections onto the feature space. The introduction of the representability vector allows us to quantitatively assess the feature space learned by different D-SSL frameworks. Similar definitions and notations have also been used in previous works studying the feature space of SSL (Wang et al., 2022; Liu et al., 2022). Based on this definition and the above problem setup, we establish the following theorem for D-SSL based on MIM pre-training.

**Theorem 4.2.** *(Representability of Distributed MIM). Consider a distributed scenario consisting of $N = \Theta(d^{\frac{1}{20}})$ clients and following the above non-iid setup with $\tau = d^{\frac{1}{5}}$ and $\mu = d^{-\frac{1}{5}}$. For distributed SSL that utilizes Masked Image Modeling (MIM) as the pre-training approach, with a high probability, the following statements hold:*

1. *Let $r_i^M = [r_{i,1}^M, \ldots, r_{i,c}^M]^\intercal$ be the local RV learned on client $i$, then we have $1 - \frac{O(d^{-\frac{2}{5}})}{2p(1-p)d^{\frac{2}{5}} + O(d^{-\frac{2}{5}})} \leq r_{i,k}^M \leq 1$, where $i \in [N] \backslash k$.*

2. *Let $\bar{r}_{Dec}^M = [\bar{r}_1^M, \ldots, \bar{r}_c^M]^\intercal$ be the global RV learned through DecL, then we have $1 - \frac{O(d^{-\frac{2}{5}})}{2p(1-p)(1-1/|\bar{A}|)d^{\frac{2}{5}} + O(d^{-\frac{2}{5}})} \leq \bar{r}_{Dec}^M \leq 1$; while for the global RV $\bar{r}_{Fed}^M = [\bar{r}_1^M, \ldots, \bar{r}_c^M]^\intercal$ learned through FL, we have $1 - \frac{O(d^{-\frac{2}{5}})}{2p(1-p)d^{\frac{2}{5}} - \Theta(d^{\frac{7}{20}}) + O(d^{-\frac{2}{5}})} \leq \bar{r}_{Fed}^M \leq 1$.*

Theorem 4.2 shows the status of the feature space learned by distributed MIM with different objectives (i.e., local vs decentralized global vs federated global). Note that for each provided representability vector, we find a unique lower bound and a shared upper bound (considering $\sum_{j=1}^d (e_k^\intercal e_j)^2 = 1$). The

distance between the lower and upper bound states how much the learned representation fluctuates in the c unit directions, $e_1, \ldots, e_c$, associated with data generation. Therefore, the smaller the distance, the less sensitive the representation space is to the non-IID distribution of local datasets on clients. In other words, the corresponding D-SSL is more robust to heterogeneity.

By a similar proof, we derive the representability vectors for D-SSL with CL pre-training as follows.

**Theorem 4.3.** *(Representability of Distributed CL). Consider the same distributed scenario in Theorem 4.2. For distributed SSL that utilizes Contrastive Learning (CL) as the pre-training approach, with a high probability, the following statements hold:*

1. *Let $r_i^C = [r_{i,1}^C, \ldots, r_{i,c}^C]^\mathsf{T}$ be the local RV, then we have $1 - \frac{O(d^{-\frac{1}{5}})}{d^{\frac{2}{5}} + O(d^{-\frac{1}{5}})} \leq r_{i,k}^C \leq 1$ and*

   $1 - \frac{O(d^{-\frac{1}{5}})}{tr(H)d^{\frac{2}{5}} + O(d^{-\frac{1}{5}})} \leq r_{i,k}^C \leq 1$ *for similar and dissimilar augmentations, respectively.*

2. *For the global RV $\bar{r}_{Dec}^C = [\bar{r}_1^C, \ldots, \bar{r}_c^C]^\mathsf{T}$ learned through DecL, we have $1 - \frac{O(d^{-\frac{1}{5}})}{(1-1/|\bar{A}|)d^{\frac{2}{5}} + O(d^{-\frac{1}{5}})} \leq \bar{r}_{Dec}^C \leq 1$ and $1 - \frac{O(d^{-\frac{1}{5}})}{tr(H)(1-1/|\bar{A}|)d^{\frac{2}{5}} + O(d^{-\frac{1}{5}})} \leq \bar{r}_{Dec}^C \leq 1$ for similar and dissimilar augmentations; while for $\bar{r}_{Fed}^C = [\bar{r}_1^C, \ldots, \bar{r}_c^C]^\mathsf{T}$ learned through FL, we have $1 - \frac{O(d^{-\frac{1}{5}})}{d^{\frac{2}{5}} - \Theta(d^{\frac{7}{20}}) + O(d^{-\frac{1}{5}})} \leq \bar{r}_{Fed}^C \leq 1$ and $1 - \frac{O(d^{-\frac{1}{5}})}{tr(H)d^{\frac{2}{5}} - \Theta(d^{\frac{7}{20}}) + O(d^{-\frac{1}{5}})} \leq \bar{r}_{Fed}^C \leq 1.$*

Theorem 4.3 demonstrates that the local and global feature spaces learned by distributed CL are distinct from those learned by distributed MIM. However, it is not obvious which feature spaces hold a smaller gap between the lower and upper bounds. To determine which type of pre-training is less sensitive to data heterogeneity, we further compare their global feature spaces learned in DecL and FL framework, respectively, and summarize the results in the following theorem.

## 4.2   MIM is inherently more Robust than CL with Heterogeneous Data

**Theorem 4.4.** *Let $s = \max_{k \in [c]} \bar{r}_k - \min_{k \in [c]} \bar{r}_k$ be the sensitivity of distributed SSL to heterogeneous data $x \in \mathbb{R}^d$, measured as the spread of the leading c coordinates of the learned global representability vector $\bar{r}$. For any network architecture, distributed SSL satisfies the following property: $\lim_{d \to \infty} [s^C > s^M]$, where $s^C$ and $s^M$ represent the sensitivities of distributed SSL adopting contrastive learning and masked image modeling as the pre-training approach, respectively.*

The main intuition for the greater robustness (or smaller sensitivity) of distributed MIM is that CL learns representations from aligning features of the positive pair generated from the original data through data augmentation, whereas MIM aligns features of the reconstructed and the raw data to learn representations. Although the applied augmentation generally does not lead to a change in data labels (Chen et al., 2020; Chen & He, 2021), the output is still a different image. In contrast, the masking operation splits the original image into the masked and unmasked parts, but a portion of the original data is retained in both parts. As a result, CL learns a local representation with greater randomness, and that additional randomness is also biased by local labels. Considering that data heterogeneity already exists among clients, the global representation learned by distributed CL is less uniform than that learned by distributed MIM.

## 4.3   Impact of the Average Client Connectivity on Non-IID Robustness

Next, we shift our focus to another dimension that distinguishes D-SSL algorithms and address the question: how does the network architecture affect the robustness of the feature space learned by D-SSL? The tool for solving this question is again the bounds of the representability vector. For the DecL setup where clients directly communicate with their direct neighbors, Theorem 4.2 and 4.3 have implicitly shown the answer.

**Corollary 4.5.** *For any SSL pre-training approaches, if the distributed scenario is fully decentralized (i.e., without a central server), the robustness of distributed SSL against heterogeneous local data improves with the average connectivity $|\bar{A}|$ between clients in the network.*

Corollary 4.5 also implies that the robustness of D-SSL conducted in a federated setup should be no worse than in a fully decentralized network. Consider the best case of the network topology, where each client can communicate with all other clients in the network. In this case, each client receives a model aggregated by the local models from all clients, which is exactly the global model distributed by the server in the federated setup. We can continue exploring to verify that this intuition is correct. Theoretically, combining Theorem 4.2, Theorem 4.3, and Corollary 4.5, we arrive at another main theorem addressing the question introduced at the beginning of this section.

**Theorem 4.6.** *For any SSL pre-training paradigms, distributed SSL satisfies the following property:* $\lim_{d \to \infty}[s_{Dec} \geq s_{Fed}]$, *where* $s_{Dec} = \max_{k \in [c]} \bar{r}_{Dec}^{(k)} - \min_{k \in [c]} \bar{r}_{Dec}^{(k)}$ *denotes the sensitivity of distributed SSL performed in the DecL setup (i.e., clients directly communicate with neighbors), and* $s_{Fed} = \max_{k \in [c]} \bar{r}_{Fed}^{(k)} - \min_{k \in [c]} \bar{r}_{Fed}^{(k)}$ *represents the sensitivity of distributed SSL performed in the FL setup (i.e., all clients are indirectly connected through the central server).*

This theorem further demonstrates the robustness trade-off between applying SSL in federated and decentralized frameworks. For less concern about the impact of data heterogeneity, we should conduct distributed SSL in a federated setup (often also referred to as federated self-supervised learning (Zhuang et al., 2021b;a; Lubana et al., 2022; Rehman et al., 2023)). However, the decentralized case is more common in reality, as it is challenging to provide a central server that can be trusted by all clients and has stable communication with them. Then, we can consider increasing the average connectivity between clients to minimize the negative impact of heterogeneous data on training (e.g., identifying under-connected clients and creating additional direct communication links).

## 5 MAR LOSS: AN ILLUSTRATIVE CASE STUDY IN ENHANCING ROBUSTNESS

The preceding analysis has addressed the main focus of this paper by establishing theoretical insights into the robustness of different D-SSL frameworks under heterogeneous data. As a further step, we illustrate how these insights can guide a more robust algorithmic design. In particular, our results show that although distributed MIM is fundamentally more robust than CL, its training dynamics are dominated by the client-specific covariance, causing local encoders to drift toward different directions before aggregation gradually mitigates this effect. This observation motivates us to refine the MIM objective with an additional term that explicitly and dynamically promotes consistency between local and global masked representations, which we term MAR loss. The integration of MAR into both federated and decentralized frameworks is summarized in Algorithm 1 and Algorithm 2

Formally, MAR loss augments the MIM objective with an alignment regularization term:

$$\mathcal{L}_{MAR} = \mathbb{E}_{x \sim D_i} \mathbb{E}_{x_1, x_2 | x} \Big[ \|f_d(f_e(x_1)) - x_2\|^2 + \gamma_t^{(i)} \cdot \text{A-MMD}(z_i, \bar{z}) \Big], \quad (5)$$

where $z_i = f_e(x_1)$ and $\bar{z}$ denote the local masked and global representations, and $\gamma_t^{(i)} > 0$ is a dynamic weight for alignment. The alignment regularizer is based on *Maximum Mean Discrepancy (MMD)*, a widely used measure of distributional discrepancy in machine learning (Gretton et al., 2012; Li et al., 2017; Gong et al., 2016). MMD compares whether two distributions $P$ and $Q$ differ by mapping samples into a reproducing kernel Hilbert space (RKHS) and evaluating differences in their feature means. Typically, MMD adopts a Gaussian kernel $k(x, x') = \exp(-\|x - x'\|^2 / 2\sigma^2)$.

In MAR, we employ an adaptive version (A-MMD) to compare the feature spaces of local and global representations more robustly. Unlike prior FL works that use vanilla MMD (Ma et al., 2024; Hu et al., 2024; Liao et al., 2024b), A-MMD selects the kernel bandwidth automatically rather than fixing it. Given batches of local and global embeddings of equal size $B$, A-MMD is computed as:

$$\text{A-MMD}(z_i, \bar{z}) = \frac{1}{B(B-1)} \left( \sum_{a \neq b} k(z_{i,a}, z_{i,b}) + \sum_{a \neq b} k(\bar{z}_a, \bar{z}_b) \right) - \frac{2}{B^2} \sum_{a=1}^{B} \sum_{b=1}^{B} k(z_{i,a}, \bar{z}_b), \quad (6)$$

with the adaptive kernel defined as $k(z, z') = exp(-\frac{\|z - z'\|}{2(\text{mean}_{a \neq b}\|z_a - z_b\|)^2})$. This data-driven choice ensures stability across non-IID clients by scaling the kernel to the observed embedding distribution.

Finally, to balance early-stage consensus and late-stage efficiency, we design the regularization weight $\gamma_t^{(i)}$ to decay smoothly from $\gamma_{\max}$ to $\gamma_{\min}$. We adopt a cosine schedule based on client participation:

$$\gamma_t^{(i)} = \gamma_{\min} + (\gamma_{\max} - \gamma_{\min}) \cdot \tfrac{1}{2} \Big( 1 + \cos \tfrac{\pi \cdot \omega_t^{(i)}}{\Omega} \Big), \quad (7)$$

where $\omega_t^{(i)}$ counts the number of times client $i$ has been selected up to round $t$, and $\Omega$ controls the decay horizon. In DecL, where all clients participate every round, one can simply set $\Omega = T$. In FL with partial participation, a practical choice is the expected number of selections per client, or $T$ as a default. This schedule applies stronger alignment when client divergence is most pronounced, and gradually relaxes toward $\gamma_{\min}$ as training progresses, ensuring dynamic robustness gains.

# 6 EXPERIMENTS

In this section, we conduct extensive experiments to validate the correctness of our derived theoretical insights and evaluate the effectiveness of the MAR loss in improving the robustness of distributed MIM against data heterogeneity. We first introduce the experimental setup. Then we assess our results in different datasets, model backbones, and distributed settings.

## 6.1 EXPERIMENTAL SETUP

**Datasets and Distributed Simulation.** We pre-train our models on the Mini-ImageNet dataset (Vinyals et al., 2016), which contains 60,000 images extracted from ImageNet (Deng et al., 2009). To simulate a distributed scenario with label non-IIDness, the dataset is partitioned by sampling the class priors of the Dirichlet distribution (Hsu et al., 2019). More heterogeneous division can be made with a smaller Dirichlet parameter $\alpha$ during sampling, while the IID case is simulated with a very large $\alpha$. Besides, we follow prior works to simulate feature heterogeneity by uniformly dividing datasets and applying unique data augmentation for each client (Wang et al., 2022; Zhu et al., 2021). Hence, the local labels are kept the same but features are skewed into different domains before training. Furthermore, to simulate DecL, we use the Erdős-Rényi model (ERDdS & R&wi, 1959) to initialize a connected network with the number of clients and the average connectivity as inputs and return the adjacency matrix $A$. For FL, we additionally assume a central server connecting with all clients. After pre-training, the models' backbones are fine-tuned on benchmark datasets, including CIFAR-10, CIFAR-100 (Krizhevsky et al., 2009), and ImageNet. The fine-tuning accuracies are used for analysis.

**Implementation Details.** For our experiments, we use ResNet (He et al., 2016) and Vision Transformer (ViT) (Dosovitskiy et al., 2020) as the model architecture. Following the problem setup in theoretical analysis, we select Simsiam (Chen & He, 2021) and MAE (He et al., 2022) as the representatives of CL and MIM pre-training, respectively. In original works, Simsiam is used to pre-train ResNet models, while MAE is used to pre-train ViTs. We implement two new SSL baselines to show that our theoretical insights apply to any model architecture. One uses Simsiam to pre-train ViTs, and the other one pre-trains ResNet through MAE. Furthermore, we follow the classical distributed algorithms, D-PSGD (Lian et al., 2017) and FedAvg (McMahan et al., 2017a), to implement the DecL and FL frameworks, and then implement our FedMAR and DecMAR algorithms based on these frameworks. All our codes are implemented in Python using the Pytorch framework and executed on a server with 4 NVIDIA® RTX 3090 GPUs. The detailed training setup and server configuration can be found in Appendix A.2. Our codes are also available at `https://github.com/xuanyuLawrence/FedMAR-DecMAR`.

## 6.2 EMPIRICAL STUDY

**Insensitivity Superiority of Distributed MIM.** Table 1 compares the impact of data heterogeneity on the pre-training effectiveness between distributed MIM and CL. With highly heterogeneous data, the learned local feature space will be significantly different across clients, resulting in a greater divergence between local and global feature space and a larger drop in the performance compared to the IID setup (Zhuang et al., 2021a;b; Lubana et al., 2022). Across various datasets and backbone architectures, we observe that distributed MIM consistently exhibits a smaller gap between IID and non-IID settings compared to distributed CL. The experimental results align with Theorem 4.4, verifying that MIM is less sensitive than CL when handling heterogeneous data in distributed scenarios. To further substantiate this theoretical insight, we also visualize the local and global feature spaces learned by distributed MIM and CL and compute the $l_2$-norm weight distance between their local and global models. Please see Appendix A.3.1 for these external experimental results.

**Impact of Average Connectivity on Non-IID Robustness**. We verify our second insight by setting up decentralized networks with different average connectivity $|\bar{A}|$. For the same $|\bar{A}|$, we consider

Table 1: **Fine-tuning accuracy (%) of backbones pre-trained by different D-SSL algorithms.** All results provided in this table are the mean of three trials (L/non-IID = Label Non-IID; F/non-IID = Feature Non-IID). The values in brackets denote the gap between IID and non-IID performance.

| | CIFAR-10 | | | CIFAR-100 | | | ImageNet | | |
|---|---|---|---|---|---|---|---|---|---|
| | IID | L/non-IID | F/non-IID | IID | L/non-IID | F/non-IID | IID | L/non-IID | F/non-IID |
| Simsiam + CNN | 86.03 | 84.33 (↓1.70) | 84.62 (↓1.41) | 58.91 | 57.80 (↓1.11) | 57.81 (↓1.10) | 46.74 | 46.10 (↓0.64) | 46.41 (↓0.33) |
| MAE + CNN | 87.28 | 86.97 (↓**0.31**) | 86.17 (↓**1.11**) | 57.86 | 57.77 (↓**0.09**) | 57.20 (↓**0.66**) | 45.88 | 45.87 (↓**0.01**) | 45.80 (↓**0.08**) |
| Simsiam + ViT | 72.32 | 69.50 (↓2.82) | 70.66 (↓1.66) | 48.60 | 43.49 (↓5.11) | 43.07 (↓5.53) | 61.97 | 59.86 (↓2.11) | 59.13 (↓2.84) |
| MAE + ViT | 69.90 | 68.20 (↓**1.70**) | 69.32 (↓**0.58**) | 50.04 | 48.95 (↓**1.09**) | 49.60 (↓**0.44**) | 62.69 | 62.25 (↓**0.44**) | 62.51 (↓**0.18**) |

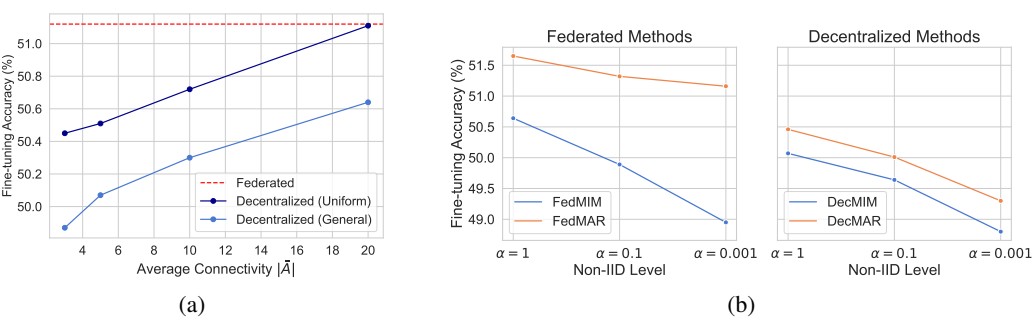

(a)                                                                 (b)

Figure 2: (a) **Impact of the average connectivity between clients on the non-IID robustness.** Models are pre-trained in a network with 20 clients and then fine-tuned on CIFAR-100. The blue line shows the results of DecL, and the orange line shows FL results. (b) **Comparison of MAR and MIM loss on robustness to data heterogeneity in federated and decentralized settings.**

two cases: (1) a general case where the number of neighbors $|A_i|$ varies across clients, and (2) a uniform case where all clients have the same connectivity, i.e., $\forall i \in [N], |A_i| = |\bar{A}|$. Additionally, we set up an FL scenario with 20 clients training in parallel per round. Figure 2a shows that Corollary 4.5 is correct. We can observe that the fine-tuning accuracy of decentralized SSL increases with $|\bar{A}|$. Moreover, Figure 2a provides empirical evidence for Theorem 4.6. We find that pre-training in FL is no less robust than in DecL against heterogeneous data. Additional results using alternative consensus matrices for DecL are given in Appendix A.3.2, and confirm the same robustness ordering.

**Effectiveness of MAR loss.** To further illustrate the practical relevance of our analysis, we evaluate MAR loss against the standard MIM objective in both FL and DecL frameworks under varying degrees of heterogeneity. Figure 2b shows that, as the level of non-IIDness increases (i.e., the Dirichlet parameter $\alpha$ decreases from 1 to 0.001), the fine-tuning accuracy of all methods declines. Nevertheless, models pre-trained with MAR loss consistently outperform those trained with the vanilla MIM loss across all non-IID levels. This trend holds in both FL and DecL settings, suggesting that MAR loss can effectively reduce the sensitivity of distributed MIM to heterogeneity.

Besides, to provide a more comprehensive evaluation, we extend the comparison to recent federated SSL baselines. We evaluate the effectiveness of our FedMAR by comparing it against several state-of-the-art (SOTA) federated self-supervised learning (F-SSL) baselines in a non-IID distributed setting. The SOTA baselines involve: **1) FedU** (Zhuang et al., 2021a): Using the divergence-aware predictor module for dynamic updates within the self-supervised BYOL network (Grill et al., 2020); **4) FedEMA** (Zhuang et al., 2021b): Employing EMA of the global model to adaptively update online networks; **5) Orchestra** (Lubana et al., 2022): Combining clustering algorithms with Federated Learning for better model aggregation. **6) FeatARC** (Wang et al., 2022): Combing clustering techniques with feature alignment; **7) LDAWA** (Rehman et al., 2023): Smartly aggregating models according to the angular divergence between local models; and **8) Fed$U^2$** (Liao et al., 2024a): Optimizing training with the flexible uniform regularizer and efficient unified aggregator. Following prior works (Zhuang et al., 2021a; Rehman et al., 2023), we simulate a highly heterogeneous scenario with 100 clients sampled from a Dirichlet distribution with $\alpha = 0.1$. In each round, 5 clients are randomly selected and each conducts 10 epochs of local training for 200 rounds in total.

Table 2: **Comparison of FedMAR with the state-of-the-art F-SSL methods on the Non-iid version ($\alpha = 0.1$) under cross-device ($n = 100$) settings.** Each method was pre-trained with Mini-ImageNet Dataset. The table shows the mean fine-tuning accuracy (%) of three trials.

| Method | Architecture | Params | GFLOPS | CIFAR-10 | CIFAR-100 | ImageNet |
|---|---|---|---|---|---|---|
| FedU | ResNet-18 | 38.47M | 7.40 | 72.02 | 38.44 | 65.10 |
| FedEMA | ResNet-18 | 38.47M | 7.40 | 70.73 | 40.78 | 65.24 |
| Orchestra | ResNet-18 | 11.84M | 7.31 | 88.87 | 70.11 | 65.02 |
| FeatARC | ResNet-18 | 11.70M | 1.83 | 89.60 | 64.11 | 68.17 |
| LDAWA | ResNet-18 | 15.39M | 1.83 | 89.95 | 68.96 | 51.43 |
| Fed$U^2$ | ResNet-18 | 15.39M | 1.83 | 82.39 | 55.49 | 45.27 |
| FedMAR(Ours) | ResNet-18 | 22.50M | 3.64 | **92.70** | **70.82** | 65.36 |
| FedMAR(Ours) | Tiny-ViT | 11.60M | 0.88 | **90.03** | **71.28** | **75.99** |

Since most baselines employ ResNet-18 (He et al., 2016) as the backbone, we first implement FedMAR with ResNet-18 for a direct comparison. As shown in Table 2, FedMAR employed on ResNet-18 achieves higher accuracy on CIFAR-10 and CIFAR-100 while obtaining comparable results on ImageNet. This indicates that MAR loss can provide tangible improvements even when using the same CNN backbone as prior methods. To further examine the generality of MAR, we also evaluate FedMAR with a lightweight Vision Transformer backbone (Tiny-ViT). Importantly, this model has a comparable number of parameters and GFLOPs to ResNet-18, ensuring fairness in comparison. In this setting, FedMAR employed on Tiny-ViT achieves superior performance on all three benchmarks, surpassing CNN-based baselines while maintaining lower computational cost. These results suggest that MAR loss is not limited to convolutional architectures and can be particularly effective when applied to transformer-based models in federated self-supervised learning.

**Extended Study on MAR.** In addition to the comparative experiments, we also conduct ablation studies on the main components of MAR loss, including the alignment term and its dynamic weighting. The detailed results of these analyses are provided in A.4.1, and A.4.2, respectively. The reported results have confirmed that each component contributes to the performance improvement led by MAR loss. Finally, we assess the practical feasibility of MAR by analyzing its privacy concerns and communication overhead. Since MAR communicates only masked embeddings, the additional communication overhead remains modest, and raw data are never shared. Privacy is therefore also preserved through masking and can be further strengthened with differential privacy. A detailed discussion is reported in Appendix A.5.

## 7 CONCLUSION

In this paper, we investigated the robustness of distributed self-supervised learning (D-SSL) under heterogeneous data. Our theoretical analysis shows that MIM-based frameworks achieve greater robustness than CL-based ones, and that the degree of robustness in decentralized learning is closely tied to the average network connectivity, with federated learning being no less robust than decentralized learning. These findings provide a principled foundation for understanding how algorithmic choices and network structures affect distributed learning with unlabeled and heterogeneous data. Beyond the theory, we also illustrated how such insights can inform practical design. As a case study, we introduced MAR loss, a refinement of the MIM objective with alignment regularization, which serves to demonstrate the applicability of our analysis. Extensive experiments across model architectures and distributed settings validate our theoretical predictions, and further confirm the utility of MAR loss in practice. We hope that our results can serve as a theoretical grounding and guiding framework for future developments in distributed self-supervised learning.

ACKNOWLEDGMENTS

This work is partly supported by the Australian Research Council Linkage Project (Grant No. LP220200893).

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

# A  APPENDIX

## A.1  FULL PSEUDOCODE OF D-SSL WITH MAR LOSS

---

**Algorithm 1** FedMAR Algorithm

---

**Input:** initial model $W^0$, number of local updates $E$, number of training rounds $T$, learning rate $\eta$, the upper bound of regularization weight $\gamma_{\max}$, the lower bound $\gamma_{\min}$
**Output:** optimized global model $W^T$
1: **for** $t = 0, \ldots, T-1$ **do**
2:     **if** $t = 0$ **then**
3:        server broadcasts $W^t$ to $\mathcal{C} \sim [N]$
4:     **else**
5:        computes $\gamma_t^{(i)}$ by $\gamma_{\max}$ and $\gamma_{\min}$ on server (shown in Eq.(7))
6:        server broadcasts $W^t, \bar{z}, \gamma_t^{(i)}$ to $\mathcal{C} \sim [N]$
7:     **end if**
8:     **for** client $i \in \mathcal{C}$ in parallel **do**
9:        $W_{i,0}^t \leftarrow W^t$
10:       **if** $t = 0$ **then**
11:         $W_{i,E}^t, z_i \leftarrow SGD(W_{i,0}^t, \eta, E, \mathcal{L}_{MIM})$
12:       **else**
13:         $W_{i,E}^t, z_i \leftarrow SGD(W_{i,0}^t, \eta, E, \mathcal{L}_{MAR}(\bar{z}, \gamma_t^{(i)}))$ (shown in Eq.(5))
14:       **end if**
15:       sends $W_{i,E}^t, z_i$ to server
16:     **end for**
17:     $\bar{z} = \frac{1}{|\mathcal{C}|} \sum_{i \in \mathcal{C}} z_i$
18:     $W^{t+1} \leftarrow \frac{1}{|\mathcal{C}|} \sum_{i \in \mathcal{C}} W_{i,E}^t$
19: **end for**

---

**Algorithm 2** DecMAR Algorithm

---

**Input:** initial models $W_{i,E}^{-1}$, number of local updates $E$, number of training rounds $T$, learning rate $\eta$, the upper bound of regularization weight $\gamma_{\max}$, the lower bound $\gamma_{\min}$
**Output:** optimized global model $W^T$
1: **for** $t = 0, \ldots, T-1$ **do**
2:     **for** client $i \in [N]$ in parallel **do**
3:        **if** $t = 0$ **then**
4:          send $W_{i,E}^{t-1}$ to its neighbors
5:        **else**
6:          computes $\gamma_t^{(i)}$ by $\gamma_{\max}$ and $\gamma_{\min}$ for each neighbor (shown in Eq.(7))
7:          send $W_{i,E}^{t-1}, z_i, \gamma_t^{(i)}$ to its neighbors
8:          $\bar{z} = \frac{1}{|A_i|} \sum_{j \in A_i} z_j$
9:        **end if**
10:       $W_{i,0}^t \leftarrow \frac{1}{|A_i|} \sum_{j \in A_i} W_{j,0}^{t-1}$
11:       **if** $t = 0$ **then**
12:         $W_{i,E}^t, z_i \leftarrow SGD(W_{i,0}^t, \eta, E, \mathcal{L}_{MIM})$
13:       **else**
14:         $W_{i,E}^t, z_i \leftarrow SGD(W_{i,0}^t, \eta, E, \mathcal{L}_{MAR}(\bar{z}, \gamma_t^{(i)}))$ (shown in Eq.(5))
15:       **end if**
16:     **end for**
17: **end for**
18: $W^T \leftarrow \frac{1}{N} \sum_{i \in [N]} W_{i,E}^{T-1}$

---

## A.2 DETAILS ABOUT EXPERIMENT SETUP

In this section, we have provided two tables to present our experiment setup. Table 3 shows the experiment details, which include the specific settings for the model architecture, dataset, scenario, and training. Table 4 demonstrates the setup of the running environment, including the configuration of our test server.

Table 3: Settings of Experiments.

|  | Details |
|---|---|
| Model Architecture | ResNet, Vision Transformer (ViT) |
| Number of layers in ResNet | 18 |
| Number of blocks in ViT | 5 |
| Pre-train Method | MAE, Simsiam |
| Pre-train Dataset | Mini-ImageNet |
| Fine-tune Dataset | CIFAR-10/100, ImageNet |
| Non-IID Options (i.e. the value of $\alpha$) | $\{1e5\ (\text{IID}), 1, 0.1, 0.01, 0.001\}$ |
| Options for the $\gamma$ used in MAR loss | $\{1, 0.1, 0.01, 0.001\}$ |
| **For Federated Learning (FL):** | |
| Number of clients | 100 |
| Number of sampled clients per round | 5 |
| Number of local training epochs | 2 |
| Number of total training rounds | 100 |
| **For Decentralized Learning (DecL):** | |
| Number of clients | 20 |
| Options for average connectivity | $3, 5, 10, 20$ (equals to FL) |
| Number of local training epochs | 1 |
| Number of total training rounds | 25 |
| Fine-tuning Epochs | 50/100 (CIFAR-10/100), 20/100 (ImageNet) |
| Pre-train Batch Size | 128 |
| Fine-tune Batch Size | 256 (CIFAR-10/100), 1024 (ImageNet) |
| Base Learning Rate | 1.5e-4 |

Table 4: Settings of Running Environment.

| Config | Details |
|---|---|
| Server GPU Count | 4 |
| Server GPU Type | RTX 3090 (24GB) |
| Server CPU Type | AMD EPYC 7282 16-Core |
| CUDA | 12.4 |
| Framework | PyTorch |

## A.3 EXTERNAL EXPERIMENTS

### A.3.1 FEATURE SPACE VISUALIZATION AND MODEL DIFFERENCE

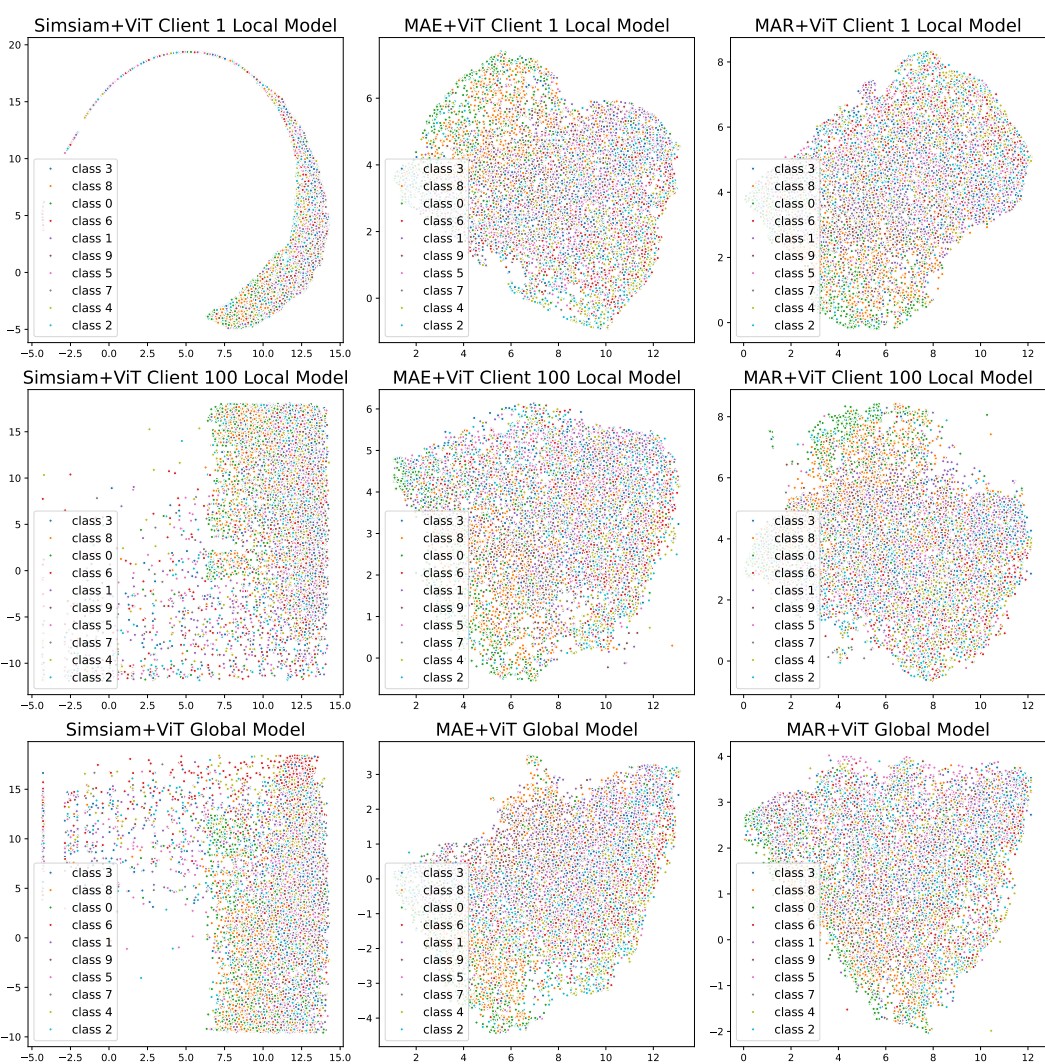

Figure 3: **Visualization of the feature space of local and global model in Non-IID setting.** Each column stands for a D-SSL framework (i.e., pre-training ViT by Simsiam, pre-training ViT by MAE, and pre-training ViT by MAR). The first row shows the local feature space from client 1, the second row shows the local feature space from client 100, and the last row shows the global feature space.

Besides Table 1 demonstrating the non-IID robustness of distributed CL and MIM by the gap in fine-tuning accuracy, we further explore the differences in their learned features empirically. Specifically, we simulate a heterogeneous setting with 100 clients using a Dirichlet sampling with $\alpha = 0.1$. For each D-SSL framework, we obtain three pre-trained ViT backbones: (1) a global model trained using FL across all clients; and (2) two local models trained solely on data from client 1 and client 100, respectively. To compare their learned feature spaces, we extract the encoder features of each model. These high-dimensional features are first projected to 20 dimensions using principal component analysis (PCA) and then embedded into 2D space using Umap (McInnes et al., 2018) for visualization.

Figure 3 presents the feature of local and global models learned by each D-SSL method. Each column corresponds to one method, while each row shows features from a specific model (client 1, client 100, and the global model). We observe that for distributed MIM methods, the local features are more aligned with each other and also closer to the global features, suggesting more consistent

representations across heterogeneous clients. In contrast, distributed CL exhibits greater divergence between local and global features, indicating that it is inherently more sensitive to data heterogeneity.

To provide a more quantitative comparison, we also show the weight differences between local and global models in Table 5. In particular, we compute the layer-wise $\ell_2$-norm difference between local and global model weights and report the sum across all layers. The results show that distributed MIM methods (MAE and MAR) yield significantly lower weight distances compared to distributed CL, reinforcing the observation that MIM leads to more stable and consistent model updates in the presence of non-IID data.

Table 5: **Weight distance between local and global models learned from different D-SSL methods.**

| $l_2$-Norm Difference | SimSiam + ViT | MAE + ViT | MAR + ViT |
|---|---|---|---|
| local 1 vs local 100 | 45.37 | 36.10 | **35.75** |
| local 1 vs global | 40.34 | 31.57 | **31.38** |
| local 100 vs global | 38.39 | 31.77 | **31.25** |

A.3.2 Effect of Different Consensus Matrices on Robustness Theory

Table 6: **CIFAR-100 Accuracy (%) of decentralized MIM under different consensus matrices.** Results are averaged over three test runs.

| Method | Avg Connectivity $\approx 5$ | Avg Connectivity $\approx 10$ |
|---|---|---|
| **FL (reference)** | **52.72** | **52.72** |
| DecL with Data Size Weights | 46.62 | 52.28 |
| DecL with Degree Normalized | 45.34 | 52.08 |
| DecL with Doubly Stochastic | 45.22 | 52.18 |
| DecL with Push Sum | 45.19 | 52.22 |

This experiment examines whether the robustness findings in Section 4.3 remain valid under different choices of consensus matrices in decentralized learning. The theoretical bounds link robustness to the average connectivity of the network graph, while the average connectivity is closely related to how efficiently information mixes across clients. If various consensus rules produced qualitatively different mixing behavior, they could in principle affect robustness. To test this, we conduct two groups of experiments on decentralized networks with 20 clients under strong non-IID conditions using $\alpha = 0.1$. In the first group, the network has an average connectivity of around 5, while in the second group, connectivity is increased to about 10. Within each group, we pre-train distributed MIM using four commonly adopted consensus schemes, including data size weighting, degree normalized averaging, doubly stochastic matrices, and push sum, and compare all results against a federated learning baseline with the same number of clients. The results in Table 6 show a consistent pattern. When connectivity is low, all decentralized variants suffer a noticeable accuracy loss relative to federated learning, and the specific consensus rule makes only minor differences. When connectivity increases, all decentralized variants recover to a level that is close to the federated baseline yet never surpass it. These findings confirm that the qualitative ordering predicted by the theory persists. The choice of consensus matrix influences only constant factors in mixing but does not overturn the robustness relation that federated learning is at least as robust as decentralized learning.

A.4 Ablation Studies on MAR

A.4.1 Ablation on Alignment Metric

Our MAR loss (Eq. 5) involves two key components: the dynamic regularization weight $\gamma_t$ and the A-MMD distributional penalty used to align local and global representations. To understand their impact, we perform ablation studies on each component. We first evaluate the contribution of the alignment metric.

For baselines, we consider two commonly used choices in prior work: cosine similarity, which has been widely adopted in federated SSL studies for enforcing alignment between local and global

Table 7: **Evaluation of different alignment metrics for MAR loss on CIFAR-100.** We report accuracy (%) under three settings of fixed $\gamma$: 1e−1, 1e−2, and 0 (degenerate to vanilla MIM).

| Metric | $\gamma = 1e{-}1$ | $\gamma = 1e{-}2$ | $\gamma = 0$ |
|---|---|---|---|
| Cosine Similarity | 51.71 | 52.47 | 51.45 |
| Vanilla MMD ($\sigma = 1$) | 51.79 | 52.12 | 51.45 |
| A-MMD (median $\sigma$) | 52.42 | 54.13 | 51.45 |
| A-MMD (mean $\sigma$) [Ours] | **54.09** | **54.39** | 51.45 |

feature spaces (Wang et al., 2022), and vanilla MMD with a fixed kernel bandwidth, which has also been explored in recent federated learning works (Ma et al., 2024; Hu et al., 2024; Liao et al., 2024b). On top of these, we evaluate our adaptive variant A-MMD, where the kernel bandwidth is chosen automatically based on either the median or mean of pairwise distances. As shown in Table 7, A-MMD consistently outperforms cosine similarity and vanilla MMD across different $\gamma$ values. Between the two adaptive variants, using the mean of pairwise distances provides slightly better performance, and we adopt this as our default design.

### A.4.2 ABLATION ON REGULARIZATION WEIGHT

Table 8: **Evaluation of regularization weight $\gamma$ for MAR loss on CIFAR-100.**

| Weight Schedule | Acc(%) |
|---|---|
| $\gamma = 1$ | 51.50 |
| $\gamma = 1e{-}1$ | 54.09 |
| $\gamma = 1e{-}2$ | 54.39 |
| $\gamma = 1e{-}3$ | 53.55 |
| $\gamma : 1e{-}1 \rightarrow 1e{-}3$ (cosine decay) | **54.91** |

Next, we analyze the impact of the regularization weight $\gamma$ by fixing the alignment metric to A-MMD. Results in Table 8 show that using a large weight ($\gamma = 1$) degrades performance, as the alignment term overwhelms the reconstruction objective. Conversely, very small weights such as $\gamma = 1e{-}3$ reduce MAR to a near-vanilla MIM objective and fail to deliver sufficient robustness gains. Moderate fixed values such as $\gamma = 1e{-}2$ and $\gamma = 1e{-}1$ yield stronger results, but still remain below our proposed dynamic schedule.

Notably, the cosine decay schedule that smoothly decreases $\gamma$ from $1e{-}1$ to $1e{-}3$ achieves the best performance (**54.91%**). This validates our intuition behind dynamic weighting: stronger alignment is most beneficial in the early stage when client divergence is high, while gradually relaxing the weight avoids excessive penalty in later stages. These findings highlight the importance of the dynamic design in MAR loss, which not only achieves higher accuracy but also improves training stability.

### A.5 DISCUSSION ON PRIVACY AND COMMUNICATION OVERHEAD OF MAR

When deploying MAR loss in practice, natural concerns arise regarding the potential privacy risks and the additional communication associated with sharing local representations. We provide both quantitative and qualitative analyses below to show that these costs remain modest and manageable.

**Privacy considerations.** The information communicated by MAR is limited to local representations $z_i = f_e(x_1)$ derived from the unmasked portion of the input. Because MIM typically adopts a high masking ratio (e.g., 75% in MAE (He et al., 2022)), most raw content remains hidden and the embedding dimensionality is substantially reduced, which mitigates potential leakage. For stronger guarantees, MAR can be further combined with standard Differential Privacy (DP) mechanisms (McMahan et al., 2017b; Wei et al., 2020) by perturbing embeddings before transmission, e.g., $z_i \leftarrow f_e(x_1) + \mathcal{N}(0, \sigma^2 I)$ with $\sigma$ calibrated to satisfy $(\epsilon, \delta)$-DP.

**Communication overhead.** In addition to the standard model updates (e.g., gradients or weights), MAR transmits compact masked embeddings computed from the unmasked portion of each input. This is the sole extra payload introduced by MAR. For instance, in the MAE (ViT-B/16) setting on ImageNet with a 75% masking ratio, each image has 196 patches, of which 49 remain visible. With

hidden size 768 and batch size 256, this yields about $49 \times 768 \times 256$ float values ($\approx 36.8 \, \text{MB}$ in float32). By contrast, a full model with 86M parameters is $\approx 328 \, \text{MB}$, so the additional cost from MAR is only $\sim 11\%$ under this configuration. Crucially, in cross-device settings where small batches are common, this extra cost decreases proportionally with the batch size: at $B{=}128$ it is $\approx 18 \, \text{MB}$ ($\sim 5\%$), at $B{=}64$ it is $\approx 9 \, \text{MB}$ ($\sim 3\%$), and at $B{=}32$ it drops to around $\sim 1\%$. These calculations indicate that the MAR-induced overhead remains acceptable in realistic deployments. Moreover, MAR is optional: when minimal communication is the overriding priority, one can simply use the standard MIM objective, whose effectiveness is explained by our theory, at zero additional cost. When a small extra cost is acceptable, MAR offers corresponding robustness gains while keeping the overhead low.

### A.6 Full proof for Theoretical Analysis

#### A.6.1 Formal Assumptions

To make the theoretical analysis fully transparent and self-contained, we first summarize here all assumptions used in deriving the main results. These assumptions complement the problem setup in Section 3 and reflect the standard modeling choices commonly adopted in theoretical studies of distributed and self-supervised learning.

**Assumption A.1.** (Communication Graph). The distributed system is modeled as a fixed and connected communication graph $\mathcal{G} = (\mathcal{V}, \mathcal{E})$ with $N = |\mathcal{V}|$ clients. Each client $i$ communicates only with its neighborhood $A_i = \{j : (i, j) \in \mathcal{E}\} \cup \{i\}$. We assume $2 \leq |A_i| \leq N$ for all $i \in [N]$. The average neighborhood size $|\bar{A}| = \frac{1}{N} \sum_{i=1}^{N} |A_i|$ is used as a measure of connectivity.

*Remark* A.2. This formulation encompasses decentralized learning on arbitrary connected topologies and federated learning as the fully connected special case (i.e., by the help of the central server, all clients can indirectly communicate with each other so there exists $|A_i| = N$ for all $i \in [N]$).

**Assumption A.3.** (Consensus Weights). During decentralized aggregation, each client $i$ forms a mixing vector $w_i = \{w_{ij}\}_{j \in A_i}$ satisfying:

- $w_{ij} > 0$ only if $j \in A_i$ (topology-respecting sparsity);

- $\sum_{j \in A_i} w_{ij} = 1$ (row-stochasticity).

*Remark* A.4. The above conditions represent the standard requirements for decentralized model aggregation: each client averages only over its local neighborhood and the mixing vector is row-stochastic. This formulation covers commonly used consensus rules in decentralized optimization, including uniform averaging (McMahan et al., 2017a), degree-normalized weights (Lian et al., 2017), and symmetric doubly-stochastic schemes (Tang et al., 2022). Our analysis relies only on these basic structural properties, while more general mixing operators could in principle be incorporated by extending the corresponding aggregation step. Exploring such extensions is an interesting direction for future work, but it is not required for the results presented here.

**Assumption A.5.** (Non-degenerate Embedding). Throughout the analysis, we focus on non-trivial stationary points of the regularized objectives, where the embedding matrix $W \in \mathbb{R}^{c \times d}$ satisfies $W \neq 0$ and $\mathrm{rank}(W) = c$.

*Remark* A.6. The trivial solution $W = 0$ does not minimize the reconstruction or alignment terms in either the MIM or CL objectives, and corresponds to a representation carrying no information. Therefore, this assumption is generally satisfied in the theoretical analysis of self-supervised learning (Liu et al., 2022; Wang et al., 2022).

**Assumption A.7.** (Independence Local Sampling.) For each client $i$, the local dataset $D_i$ consists of $|D_i|$ independent samples drawn from its local distribution $\mathcal{D}_i$.

*Remark* A.8. The independence assumption is the minimal condition required for high-probability spectral norm bounds of empirical covariance matrices. It does not alter the established non-IID structure across clients, but ensures that the empirical covariance on each client concentrates around its population counterpart. This is a standard assumption in the theoretical analysis of distributed learning (Wang et al., 2022; Tang et al., 2022).

**Assumption A.9.** (Dissimilar Image Transformation for CL). When the two augmented views $(g_a(x), g_b(x))$ used in CL are generated from dissimilar transformations, we model $g_b(x)$ by a linear operator $H \in \mathbb{R}^{d \times d}$ acting on the input space. In the theoretical analysis, $H$ enters only through the quadratic form $x^\top H x$, and therefore only its symmetric component $H_{\mathrm{sym}} = (H + H^\top)/2$ is relevant. Let $\mathcal{S} = \mathrm{span}\{e_1, \ldots, e_c\}$ denote the class-dependent subspace in the non-IID generative model, and let $P$ be the orthogonal projection onto $\mathcal{S}$. We assume that

$$\mathrm{tr}(P H_{\mathrm{sym}} P) > 0.$$

*Remark* A.10. The above condition ensures that the transformation $H$ preserves nontrivial energy on the class-dependent semantic subspace $\mathcal{S}$. It is a mild requirement and is common to hold in standard contrastive learning augmentations (Chen et al., 2020; Chen & He, 2021), including rotations, flips, translations, crops, blurs, and color jittering. These transformations perturb the input in ways that do not cancel class-discriminative directions, so $\mathrm{tr}(P H_{\mathrm{sym}} P)$ remains strictly positive in practice.

### A.6.2 LEARNED REPRESENTABILITY FOR DECENTRALIZED MIM

This section provides the full proof of Theorem 4.2.

*Proof.* We begin by formulating the representability of local representation. Then, we derive the global representation based on the local feature. Since FL is different from decentralized learning in the updates, we establish the global representation for each distributed framework, respectively.

**For local feature space.** According to the alignment-style loss function of MIM shown in Eq.(4) and by the definition of Kronecker product, we have

$$\mathcal{L}_{MIM} = -\mathbb{E}[|(W(x \odot m))^\mathsf{T}(W(x \odot (1-m)))] + \frac{1}{2}||W^\mathsf{T}W||_F^2$$
$$= -\mathbb{E}[(W(\text{diag}(\text{vec}(x)) \cdot \text{vec}(m)))^\mathsf{T}(W(\text{diag}(\text{vec}(x)) \cdot \text{vec}(1-m)))] + \frac{1}{2}||W^\mathsf{T}W||_F^2. \tag{8}$$

Define
$$a = \text{diag}(\text{vec}(x)) \cdot \text{vec}(m), \qquad b = \text{diag}(\text{vec}(x)) \cdot \text{vec}(1-m), \tag{9}$$
so that the above loss becomes
$$\mathcal{L}_{MIM} = -\mathbb{E}[a^\mathsf{T}W^\mathsf{T}Wb] + \frac{1}{2}||W^\mathsf{T}W||_F^2. \tag{10}$$

Using the fact that
$$a^\mathsf{T}W^\mathsf{T}Wb = \text{tr}(a^\mathsf{T}W^\mathsf{T}Wb) = \text{tr}(W^\mathsf{T}Wba^\mathsf{T}) = \text{tr}(Wba^\mathsf{T}W^\mathsf{T}), \tag{11}$$

we obtain
$$\frac{\partial}{\partial W}(a^\mathsf{T}W^\mathsf{T}Wb) = \frac{\partial}{\partial W}(\text{tr}(Wba^\mathsf{T}W^\mathsf{T})) = W(ba^\mathsf{T} + ab^\mathsf{T}). \tag{12}$$

Together with
$$\frac{\partial(\frac{1}{2}||W^\top W||_F^2)}{\partial W} = 2WW^\mathsf{T}W, \tag{13}$$

the gradient of the complete objective becomes
$$\frac{\partial \mathcal{L}_{\text{MIM}}}{\partial W} = -W\mathbb{E}[ba^\mathsf{T} + ab^\mathsf{T}] + 2WW^\mathsf{T}W. \tag{14}$$

Setting the gradient to zero yields
$$W\mathbb{E}[ba^\mathsf{T} + ab^\mathsf{T}] = 2WW^\mathsf{T}W. \tag{15}$$

Under Assumption A.5, multiplying both sides on the left by the Moore–Penrose pseudoinverse $W^+$ reduces Eq.(15) to the stationary condition
$$\frac{1}{2}\mathbb{E}[ba^\mathsf{T} + ab^\mathsf{T}] = W^\mathsf{T}W. \tag{16}$$

Let $X_i^M$ represent the left-hand side of this equation. Consider the binary matrix $m$ used for masking is sampled uniformly from the binomial distribution with a probability $p$, we establish

$$X_i^M = \frac{1}{2}\mathbb{E}[\text{diag}(\text{vec}(x))\text{vec}(1-m)\text{vec}(m)^\mathsf{T}\text{diag}(\text{vec}(x))^\mathsf{T}$$
$$+ \text{diag}(\text{vec}(x))\text{vec}(m)\text{vec}(1-m)^\mathsf{T}\text{diag}(\text{vec}(x))^\mathsf{T}]$$
$$= \frac{2p(1-p)}{|D_i|}\sum_{j=1}^{|D_i|}(\text{diag}(\text{vec}(x_{i,j}))\text{diag}(\text{vec}(x_{i,j}))^\mathsf{T}), \tag{17}$$

where $\mathbb{E}_{x \sim D_i}[xx^\mathsf{T}] = \frac{1}{|D_i|}\sum_{j=1}^{|D_i|}(\text{diag}(\text{vec}(x_{i,j}))\text{diag}(\text{vec}(x_{i,j}))^\mathsf{T})$ denotes the empirical covariance matrix for the learning with local dataset on client $i$. Based on the setup of data generation in Section 3, we also derive the following expectation of $X_i^M$ with $\tau = d^{\frac{1}{5}}$ and $\mu = d^{-\frac{1}{5}}$:

$$\mathbb{E}\left[X_i^M\right] = \text{diag}$$

$$\left( \underbrace{2p\left(1-p\right)\tau^2 + O\left(d^{-\frac{2}{5}}\right), ..., \underbrace{2p\left(1-p\right) + O\left(d^{-\frac{2}{5}}\right)}_{i^{th}\ \text{term}}, ...,\ 2p\left(1-p\right)\tau^2 + O\left(d^{-\frac{2}{5}}\right),}_{N\ \text{terms}} \right.$$

$$\left. \underbrace{O\left(d^{-\frac{2}{5}}\right), ..., O\left(d^{-\frac{2}{5}}\right)}_{d-N\ \text{terms}} \right) \tag{18}$$

$$= \text{diag}\left( 2p\left(1-p\right)d^{\frac{2}{5}} + O\left(d^{-\frac{2}{5}}\right), ..., 2p\left(1-p\right) + O\left(d^{-\frac{2}{5}}\right), \right.$$

$$\left. ..., 2p\left(1-p\right)d^{\frac{2}{5}} + O\left(d^{-\frac{2}{5}}\right), ..., O\left(d^{-\frac{2}{5}}\right) \right)$$

Next, consider the fact that up to a positive scaling and an additive constant, the regularized MIM objective can be rewritten as the Frobenius-norm objective $\mathcal{L}(W) = \|X_i^M - W^\mathsf{T}W\|_F^2$. Thus, minimizing $\mathcal{L}_{\text{MIM}}$ solves the Frobenius-norm best rank-$c$ approximation problem for $X_i^M$. According to the Eckart-Young-Mirsky theorem (Eckart & Young, 1936), we notice that the row span of the optimal $W \in \mathbb{R}^{c \times d}$ is the span of the eigenvectors corresponding to the first $c$ eigenvalues of $X_i^M$. Denoting the set of orthonormal eigenvectors of $X_i^M$ as $\left\{v_{i,1}^M, ..., v_{i,d}^M\right\}$, we have $X_i^M = \sum_{j=1}^d \lambda_{i,j} v_{i,j}^M (v_{i,j}^M)^\mathsf{T}$, where $\lambda_{i,j} := \lambda_j(X_i^M)$ is the j-th largest eigenvalue of $X_i^M$. Therefore, the inequality below is satisfied:

$$e_k^\mathsf{T} X_i^M e_k = e_k^\mathsf{T}\left( \sum_{j=1}^d \lambda_{i,j} v_{i,j}^M (v_{i,j}^M)^\mathsf{T} \right) e_k$$

$$= \sum_{j=1}^d \lambda_{i,j} (e_k^\mathsf{T} v_{i,j}^M)^2 \tag{19}$$

$$\leq \lambda_{i,1}^M \sum_{j=1}^d (e_k^\mathsf{T} v_{i,j}^M)^2,$$

for any $e_k$ with $k \in [N] \setminus \{i\}$. On the other hand, under the data construction described in Section 3.2, the number of samples on each client equals to the sum of the samples from frequent classes and the rare class. Since each of the two frequent classes grows in polynomials of $d$, while the amount of data from the rare class is $O(d^\alpha)$ with $\alpha \in (0, 1)$, the local sample size satisfies $|D_i| = \Theta(d^\beta)$ with $\beta \geq 1$. Based on this sufficiently large sample size and Assumption A.7, the matrix concentration bounds (Vershynin, 2018) implies that the spectral norm satisfies $\|X_i^M - \mathbb{E}\left[X_i^M\right]\|_2 \leq O\left(d^{-\frac{2}{5}}\right)$ with probability at least $1 - \frac{1}{2}e^{-d^{\frac{1}{10}}}$. Building on Weyl's inequality, we obtain that with high probability,

$$\left|\lambda_{i,k}^M - \lambda_k \mathbb{E}\left[X_i^M\right]\right| \leq \|X_i^M - \mathbb{E}\left[X_i^M\right]\|_2 \leq O\left(d^{-\frac{2}{5}}\right). \tag{20}$$

By combining Eqs.(18), (19) and (20), we can derive the below lower bound for $e_k^\mathsf{T} X_i^M e_k$:

$$e_k^\mathsf{T} X_i^M e_k = e_k^\mathsf{T} \mathbb{E}\left[X_i^M\right] e_k + e_k^\mathsf{T}\left[X_i^M - \mathbb{E}\left[X_i^M\right]\right] e_k$$

$$\geq 2p\left(1-p\right)d^{\frac{2}{5}} + O\left(d^{-\frac{2}{5}}\right) - \|X_i^M - \mathbb{E}\left[X_i^M\right]\| \tag{21}$$

$$\geq 2p\left(1-p\right)d^{\frac{2}{5}} - O\left(d^{-\frac{2}{5}}\right),$$

which is led by the fact that $\|X\|_{\max} \leq \|X\|$ for symmetric $X$. Likewise, we prove the upper bound as follows:

$$
\begin{aligned}
e_k^\mathsf{T} X_i^M e_k &= e_k^\mathsf{T} \mathbb{E}\left[X_i^M\right] e_k + e_k^\mathsf{T}\left[X_i^M - \mathbb{E}\left[X_i^M\right]\right] e_k \\
&\leq 2p\left(1-p\right) d^{\frac{2}{5}} + O\left(d^{-\frac{2}{5}}\right) + \|X_i^M - \mathbb{E}\left[X_i^M\right]\| \\
&\leq 2p\left(1-p\right) d^{\frac{2}{5}} + O\left(d^{-\frac{2}{5}}\right).
\end{aligned}
\tag{22}
$$

Moreover, we notice from Eqs.(18) and (19) that the below statements hold for $\lambda_{i,1}^M$:

$$
\begin{aligned}
\lambda_{i,1}^M &\geq \lambda_1\left(\mathbb{E}\left[X_i^M\right]\right) - O\left(d^{-\frac{2}{5}}\right) \geq 2p\left(1-p\right) d^{\frac{2}{5}} - O\left(d^{-\frac{2}{5}}\right) \\
\lambda_{i,1}^M &\leq \lambda_1\left(\mathbb{E}\left[X_i^M\right]\right) + O\left(d^{-\frac{2}{5}}\right) = 2p\left(1-p\right) d^{\frac{2}{5}} + O\left(d^{-\frac{2}{5}}\right).
\end{aligned}
\tag{23}
$$

With Eqs.(21) - (23), we further establish

$$
\begin{aligned}
\sum_{j=1}^d (e_k^\mathsf{T} v_j^M)^2 &\geq \frac{2p\left(1-p\right) d^{\frac{2}{5}} - O\left(d^{-\frac{2}{5}}\right)}{2p\left(1-p\right) d^{\frac{2}{5}} + O\left(d^{-\frac{2}{5}}\right)} \\
&= \frac{2p\left(1-p\right) d^{\frac{2}{5}} + O\left(d^{-\frac{2}{5}}\right)}{2p\left(1-p\right) d^{\frac{2}{5}} + O\left(d^{-\frac{2}{5}}\right)} - \frac{2O\left(d^{-\frac{2}{5}}\right)}{2p\left(1-p\right) d^{\frac{2}{5}} + O\left(d^{-\frac{2}{5}}\right)} \\
&= 1 - \frac{O\left(d^{-\frac{2}{5}}\right)}{2p\left(1-p\right) d^{\frac{2}{5}} + O\left(d^{-\frac{2}{5}}\right)}.
\end{aligned}
\tag{24}
$$

This completes the proof for local representation.

**For global feature space.** Since the local goal can be equivalently re-formulated as $\|X_i^M - W^\mathsf{T} W\|_F^2$, by Assumptions A.1 and A.3, we re-write the global goal of D-SSL for DecL framework (shown in Eq.(1)) as

$$
\min_W \frac{1}{N} \sum_{i \in [N]} \frac{1}{|A_i|} \sum_{j \in A_i} \|X_j^M - W^\mathsf{T} W\|_F^2.
\tag{25}
$$

Note that the following function holds the same minimizer as Eq.(25):

$$
\begin{aligned}
\min_W \| &\frac{1}{N} \sum_{i \in [N]} \frac{1}{|A_i|} \sum_{j \in A_i} X_j^M - W^\mathsf{T} W\|_F^2 \\
&= \min_W \|\frac{1}{N} \sum_{i \in [N]} \overline{X_i^M} - W^\mathsf{T} W\|_F^2 \\
&= \min_W \|\overline{X^M} - W^\mathsf{T} W\|_F^2,
\end{aligned}
\tag{26}
$$

where $\overline{X_i^M} = \sum_{j \in A_i} \frac{1}{|A_i|} X_j^M$ denotes the empirical covariance matrix for training with the local datasets across the local datasets on client $i$ and its neighbors. So, finding the optimal $W$ for DecL is equivalent to solving Eq.(26). Following the derivation of Eq.(18) and linearity of expectation, we

establish

$$
\mathbb{E}\left(\overline{X_i^M}\right) = \text{diag}
$$

$$
\left( ..., \underbrace{2p\left(1-p\right)\left(\left(1-\frac{1}{|A_i|}\right)d^{\frac{2}{5}} + \frac{1}{|A_i|}\right) + O\left(d^{-\frac{2}{5}}\right)}_{j \in A_i \setminus i}, ..., \underbrace{2p\left(1-p\right)d^{\frac{2}{5}} + O\left(d^{-\frac{2}{5}}\right)}_{i^{th}\,\text{term}}, ..., \right. \tag{27}
$$

$$
\left. \underbrace{O\left(d^{-\frac{2}{5}}\right), ..., O\left(d^{-\frac{2}{5}}\right)}_{d-N\,\text{terms}} \right),
$$

where we prove with the fact that

$$
\frac{\left(|A_i|-1\right)2p\left(1-p\right)d^{\frac{2}{5}} + 2p\left(1-p\right) + |A_i| O\left(d^{-\frac{2}{5}}\right)}{|A_i|}
$$
$$
= \frac{\left(|A_i|-1\right)2p\left(1-p\right)d^{\frac{2}{5}} + 2p\left(1-p\right)}{|A_i|} + O\left(d^{-\frac{2}{5}}\right) \tag{28}
$$
$$
= 2p\left(1-p\right)\left(1-\frac{1}{|A_i|}\right)d^{\frac{2}{5}} + 2p\left(1-p\right)\frac{1}{|A_i|} + O\left(d^{-\frac{2}{5}}\right)
$$
$$
= 2p\left(1-p\right)\left(\left(1-\frac{1}{|A_i|}\right)d^{\frac{2}{5}} + \frac{1}{|A_i|}\right) + O\left(d^{-\frac{2}{5}}\right).
$$

With Eq.(27), we can also have

$$
\mathbb{E}\left(\overline{X^M}\right) = \text{diag}
$$

$$
\left( \underbrace{2p\left(1-p\right)\left(1-\frac{1}{|\bar{A}|}\right)d^{\frac{2}{5}} + O\left(d^{-\frac{9}{20}}\right), ..., 2p\left(1-p\right)\left(1-\frac{1}{|\bar{A}|}\right)d^{\frac{2}{5}} + O\left(d^{-\frac{9}{20}}\right)}_{N\,terms}, \right. \tag{29}
$$

$$
\left. ..., O\left(d^{-\frac{2}{5}}\right) \right)
$$

where we consider $\frac{1}{N}\sum_{i=1}^{N}\frac{1}{|A_i|} = |\bar{A}|$ and the fact that

$$
\frac{\sum_{i=1}^{N}\left(2p\left(1-p\right)\left(\left(1-\frac{1}{|A_i|}\right)d^{\frac{2}{5}} + \frac{1}{|A_i|}\right) + O\left(d^{-\frac{2}{5}}\right)\right)}{N}
$$
$$
= 2p\left(1-p\right)\left(\left(1-\frac{1}{N}\sum_{i=1}^{N}\frac{1}{|A_i|}\right)d^{\frac{2}{5}} + \frac{1}{N}\sum_{i=1}^{N}\frac{1}{|A_i|}\right) + O\left(d^{-\frac{2}{5}}\right)
$$
$$
= 2p\left(1-p\right)\left(\left(1-\frac{1}{|\bar{A}|}\right)d^{\frac{2}{5}} + \frac{1}{|\bar{A}|}\right) + O\left(d^{-\frac{2}{6}}\right) \tag{30}
$$
$$
= 2p\left(1-p\right)\left(1-\frac{1}{|\bar{A}|}\right)d^{\frac{2}{5}} + O\left(d^{-\frac{2}{5}}\right).
$$

Through similar proof from Eq.(21) to Eq.(23), we prove that the following statements hold for all $i \in [N]$:

$$
e_k^{\mathsf{T}}\overline{X^M}e_k \geq 2p\left(1-p\right)\left(1-\frac{1}{|\bar{A}|}\right)d^{\frac{2}{5}} - O\left(d^{-\frac{2}{5}}\right)
$$

$$
e_k^{\mathsf{T}}\overline{X^M}e_k \leq 2p\left(1-p\right)\left(1-\frac{1}{|\bar{A}|}\right)d^{\frac{2}{5}} + O\left(d^{-\frac{2}{5}}\right) \tag{31}
$$

$$\lambda_{i,1}^M \geq \lambda_1\left(\mathbb{E}\left[\overline{X^M}\right]\right) + O\left(d^{-\frac{2}{5}}\right) = 2p\left(1-p\right)\left(1 - \frac{1}{|\bar{A}|}\right)d^{\frac{2}{5}} - O\left(d^{-\frac{2}{5}}\right)$$

$$\lambda_{i,1}^M \leq \lambda_1\left(\mathbb{E}\left[\overline{X^M}\right]\right) + O\left(d^{-\frac{2}{5}}\right) = 2p\left(1-p\right)\left(1 - \frac{1}{|\bar{A}|}\right)d^{\frac{2}{5}} + O\left(d^{-\frac{2}{5}}\right),$$

(32)

which then implies:

$$\sum_{j=1}^d (e_k^\intercal \bar{v}_j^M)^2 \geq \frac{2p\left(1-p\right)\left(1 - \frac{1}{|\bar{A}|}\right)d^{\frac{2}{5}} - O\left(d^{-\frac{2}{5}}\right)}{2p\left(1-p\right)\left(1 - \frac{1}{|\bar{A}|}\right)d^{\frac{2}{5}} + O\left(d^{-\frac{2}{5}}\right)}$$

$$= \frac{2p\left(1-p\right)\left(1 - \frac{1}{|\bar{A}|}\right)d^{\frac{2}{5}} + O\left(d^{-\frac{2}{5}}\right)}{2p\left(1-p\right)\left(1 - \frac{1}{|\bar{A}|}\right)d^{\frac{2}{5}} + O\left(d^{-\frac{2}{5}}\right)} - \frac{2O\left(d^{-\frac{2}{5}}\right)}{2p\left(1-p\right)\left(1 - \frac{1}{|\bar{A}|}\right)d^{\frac{2}{5}} + O\left(d^{-\frac{2}{5}}\right)}$$

(33)

$$= 1 - \frac{O\left(d^{-\frac{2}{5}}\right)}{2p\left(1-p\right)\left(1 - \frac{1}{|\bar{A}|}\right)d^{\frac{2}{5}} + O\left(d^{-\frac{2}{5}}\right)}.$$

The proof for the global featured space learned in the decentralized learning framework has been completed. Next, consider federated learning (FL) as a special case of decentralized learning with $\forall i \in [N], |A_i| = N$. The global of FL is thus:

$$\min_W \frac{1}{N} \sum_{i \in [N]} \|X_i^M - W^\intercal W\|_F^2. \tag{34}$$

This is similar to solving

$$\min_W \|\overline{X^M} - W^\intercal W\|_F^2, \tag{35}$$

where $\overline{X^M} := \frac{1}{N}\sum_{i\in[N]} X_i^M$ denotes the empirical covariance matrix for learning with the global dataset. Then, we derive

$$\mathbb{E}\left(\overline{X^M}\right) = \text{diag}\left(2p\left(1-p\right)d^{\frac{2}{5}} - \Theta\left(d^{\frac{7}{20}}\right) + O\left(d^{-\frac{2}{5}}\right), ...,\right.$$

$$\left. 2p\left(1-p\right)d^{\frac{2}{5}} - \Theta\left(d^{\frac{7}{20}}\right) + O\left(d^{-\frac{2}{5}}\right), ...O\left(d^{-\frac{2}{5}}\right)\right)$$

(36)

where we adopt $N = \Theta(d^{\frac{1}{20}})$ have used the fact that

$$\frac{(N-1)\,2p\left(1-p\right)d^{\frac{2}{5}} + 2p\left(1-p\right) + NO\left(d^{-\frac{2}{5}}\right)}{N}$$

$$= \frac{\left(\Theta\left(d^{\frac{1}{20}}\right) - 1\right)2p\left(1-p\right)d^{\frac{2}{5}} + 2p\left(1-p\right)}{\Theta\left(d^{\frac{1}{20}}\right)} + O\left(d^{-\frac{2}{5}}\right)$$

(37)

$$= 2p\left(1-p\right)\left(1 - \Theta\left(d^{-\frac{1}{20}}\right)\right)d^{\frac{2}{5}} + \Theta\left(d^{-\frac{1}{20}}\right) + O\left(d^{-\frac{2}{5}}\right)$$

$$= 2p\left(1-p\right)d^{\frac{2}{5}} - \Theta\left(d^{\frac{7}{20}}\right) + O\left(d^{-\frac{2}{5}}\right).$$

Again, by similar arguments from Eq.(21) to Eq.(23), we further prove

$$
\begin{aligned}
\sum_{j=1}^{d} (e_k^{\mathsf{T}} \bar{v}_j^M)^2 &\geq \frac{2p\,(1-p)\,d^{\frac{2}{5}} - \Theta\left(d^{\frac{7}{20}}\right) - O\left(d^{-\frac{2}{5}}\right)}{2p\,(1-p)\,d^{\frac{2}{5}} - \Theta\left(d^{\frac{7}{20}}\right) + O\left(d^{-\frac{2}{5}}\right)} \\
&= \frac{2p\,(1-p)\,d^{\frac{2}{5}} - \Theta\left(d^{\frac{7}{20}}\right) + O\left(d^{-\frac{2}{5}}\right)}{2p\,(1-p)\,d^{\frac{2}{5}} - \Theta\left(d^{\frac{7}{20}}\right) + O\left(d^{-\frac{2}{5}}\right)} - \frac{2O\left(d^{-\frac{2}{5}}\right)}{p\,(1-p)\,d^{\frac{2}{5}} - \Theta\left(d^{\frac{7}{20}}\right) + O\left(d^{-\frac{2}{5}}\right)} \\
&= 1 - \frac{O\left(d^{-\frac{2}{5}}\right)}{2p\,(1-p)\,d^{\frac{2}{5}} - \Theta\left(d^{\frac{7}{20}}\right) + O\left(d^{-\frac{2}{5}}\right)},
\end{aligned}
\tag{38}
$$

which completes the proof of this theorem.

$\square$

### A.6.3 LEARNED REPRESENTABILITY FOR DECENTRALIZED CONTRASTIVE LEARNING

This section provides the full proof of Theorem 4.3.

**Lemma A.11.** *(Representability of Distributed CL under Similar Augmentations). Consider the same distributed scenario in Theorem 4.2. For distributed SSL that utilizes Contrastive Learning (CL) in pre-training and generate positive pairs through similar augmentations, with a high probability, the following statements hold:*

1. *Let $r_i^C = [r_{i,1}^C, \ldots, r_{i,c}^C]^\mathsf{T}$ be the local RV learned on client $i$. If positive pairs are generated by similar augmentations, we have $1 - \frac{O(d^{-\frac{1}{5}})}{d^{\frac{2}{5}} + O(d^{-\frac{1}{5}})} \leq r_{i,k}^C \leq 1$, where $i \in [N] \backslash k$.*

2. *Let $\bar{r}_{Dec}^C = [\bar{r}_1^C, \ldots, \bar{r}_c^C]^\mathsf{T}$ be the RV learned through the global objective of DecL framework, then we have $1 - \frac{O(d^{-\frac{1}{5}})}{(1-\frac{1}{|A|})d^{\frac{2}{5}} + O(d^{-\frac{1}{5}})} \leq \bar{r}^C \leq 1$.*

3. *Let $\bar{r}_{Fed}^M = [\bar{r}_1^M, \ldots, \bar{r}_c^M]^\mathsf{T}$ be the RV learned through the global objective of FL framework, we have $1 - \frac{O(d^{-\frac{1}{5}})}{d^{\frac{2}{5}} - \Theta(d^{\frac{7}{20}}) + O(d^{-\frac{1}{5}})} \leq \bar{r}_{Fed}^C \leq 1$.*

*Proof.* Following the proof in A.6.2, we first discuss local representability learned by distributed contrastive learning and then derive the global representation based on these local features. Since federated learning differs from decentralized learning in terms of updates, we construct separate global representations for each distributed framework.

**For local feature space.** Let $a = x + \xi$ and $b = x + \xi'$. Based on the loss function of contrastive learning (CL) shown in Section 3.2, we obtain

$$
\begin{aligned}
\mathcal{L}_{CL} &= -\mathbb{E}_{x \sim D_i}[(W(x+\xi))^\mathsf{T}(W(x+\xi'))] + \frac{1}{2}||W^\mathsf{T}W||_F^2 \\
&= -\mathbb{E}_{x \sim D_i}[a^\mathsf{T}W^\mathsf{T}Wb] + \frac{1}{2}||W^\mathsf{T}W||_F^2.
\end{aligned}
\tag{39}
$$

By the same derivation between Eq.(11)-Eq.(14), the gradient of the above function is

$$
\frac{\partial \mathcal{L}_{\mathrm{CL}}}{\partial W} = -W\mathbb{E}\big[(x+\xi')(x+\xi)^\mathsf{T} + (x+\xi)(x+\xi')^\mathsf{T}\big] + 2WW^\mathsf{T}W.
\tag{40}
$$

To find the minimizer of $\mathcal{L}_{CL}$, we solve for

$$
-W\mathbb{E}\big[(x+\xi')(x+\xi)^\mathsf{T} + (x+\xi)(x+\xi')^\mathsf{T}\big] + 2WW^\mathsf{T}W = 0.
\tag{41}
$$

Under Assumption A.5, it leads to

$$
\frac{1}{2}\mathbb{E}\big[2xx^\mathsf{T} + x\xi^\mathsf{T} + \xi'x^\mathsf{T} + x(\xi')^\mathsf{T} + \xi x^\mathsf{T} + \xi'\xi^\mathsf{T} + \xi(\xi')^\mathsf{T}\big] = W^\mathsf{T}W.
\tag{42}
$$

Similarly, let $X_i^C$ represent the left-hand side of this equation. We can then establish

$$
X_i^C = \frac{1}{2|D_i|}\sum_{j=1}^{|D_i|}\big(2x_{i,j}x_{i,j}^\mathsf{T} + x_{i,j}\xi_{i,j}^\mathsf{T} + \xi_{i,j}'x_{i,j}^\mathsf{T} + x_{i,j}(\xi')_{i,j}^\mathsf{T} + \xi_{i,j}x_{i,j}^\mathsf{T} + \xi_{i,j}'\xi_{i,j}^\mathsf{T} + \xi_{i,j}(\xi')_{i,j}^\mathsf{T}\big),
\tag{43}
$$

where $X_i^C$ represents the empirical covariance matrix for the local feature learned by CL on client $i$. Considering that $\xi, \xi' \sim \mathcal{N}(0, I)$, we also derive the following expectation of $X_i^C$:

$$\mathbb{E}\left[X_i^C\right] = \mathrm{diag}$$

$$\left(\tau^2 + O\left(d^{-\frac{2}{5}}\right) + 2O\left(d^{-\frac{1}{5}}\right), ..., \underbrace{1 + O\left(d^{-\frac{2}{5}}\right) + 2O\left(d^{-\frac{1}{5}}\right)}_{i^{\text{th}} \text{ term}}, ..., \tau^2 + O\left(d^{-\frac{2}{5}}\right) + 2O\left(d^{-\frac{1}{5}}\right),\right.$$
$$\underbrace{\phantom{\tau^2 + O\left(d^{-\frac{2}{5}}\right)}}_{N \text{ terms}}$$

$$\left....\underbrace{2O\left(d^{-\frac{1}{5}}\right) + O\left(d^{-\frac{2}{5}}\right), ..., 2O\left(d^{-\frac{1}{5}}\right) + O\left(d^{-\frac{2}{5}}\right)}_{d-N \text{ terms}}\right)$$

$$= \mathrm{diag}\left(d^{\frac{2}{5}} + O\left(d^{-\frac{1}{5}}\right), ..., 1 + O\left(d^{-\frac{1}{5}}\right), ..., d^{\frac{2}{5}} + O\left(d^{-\frac{1}{5}}\right), ..., O\left(d^{-\frac{1}{5}}\right)\right) \tag{44}$$

Next, using similar arguments from Eqs. (19) to (23), we arrive at the below results:

$$d^{\frac{2}{5}} - O\left(d^{-\frac{1}{5}}\right) \le e_k^\intercal X_i^C e_k \le d^{\frac{2}{5}} + O\left(d^{-\frac{1}{5}}\right)$$
$$d^{\frac{2}{5}} - O\left(d^{-\frac{1}{5}}\right) \le \lambda_{i,1}^C \le d^{\frac{2}{5}} + O\left(d^{-\frac{1}{5}}\right). \tag{45}$$

With these inequalities, we derive

$$\sum_{j=1}^d (e_k^\intercal v_j^C)^2 \ge \frac{d^{\frac{2}{5}} - O\left(d^{-\frac{1}{5}}\right)}{d^{\frac{2}{5}} + O\left(d^{-\frac{1}{5}}\right)}$$
$$= 1 - \frac{O\left(d^{-\frac{1}{5}}\right)}{d^{\frac{2}{5}} + O\left(d^{-\frac{1}{5}}\right)}, \tag{46}$$

which completes the proof of the local part.

**For global feature space.** Since the local goal can be equivalently reformulated as $\|X_i^C - W^\intercal W\|_F^2$, by Assumptions A.1 and A.3, the global goal of distributed contrastive learning in the decentralized learning (DecL) framework is given by

$$\min_W \sum_{i \in [N]} \frac{1}{N} \sum_{j \in A_i} \frac{1}{|A_i|} \|X_j^C - W^\intercal W\|_F^2. \tag{47}$$

Furthermore, we find this is equivalent to solving

$$\min_W \|\frac{1}{N} \sum_{i \in [N]} \frac{1}{|A_i|} \sum_{j \in A_i} X_j^C - W^\intercal W\|_F^2$$
$$= \min_W \|\frac{1}{N} \sum_{i \in [N]} \overline{X_i^C} - W^\intercal W\|_F^2 \tag{48}$$
$$= \min_W \|\overline{X^C} - W^\intercal W\|_F^2.$$

Again, using similar arguments from Eq. (27) to Eq. (33), we further establish

$$
\sum_{k=1}^{d} (e_k^\mathsf{T} \bar{v}_j^C)^2 \geq \frac{\left(1 - \frac{1}{|\bar{A}|}\right) d^{\frac{2}{5}} - O\left(d^{-\frac{1}{5}}\right)}{\left(1 - \frac{1}{|\bar{A}|}\right) d^{\frac{2}{5}} + O\left(d^{-\frac{1}{5}}\right)}
$$
$$
= 1 - \frac{O\left(d^{-\frac{1}{5}}\right)}{\left(1 - \frac{1}{|\bar{A}|}\right) d^{\frac{2}{5}} + O\left(d^{-\frac{1}{5}}\right)}.
\tag{49}
$$

The proof for the global feature space learned in the DecL framework has been completed. Next, denote federated learning (FL) as a special case of decentralized learning with $\forall i, |A_i| = N$. The global objective of FL is expressed as

$$
\min_W \|\overline{X^C} - W^\mathsf{T} W\|_F^2.
\tag{50}
$$

where we denote $\overline{X^C} := \frac{1}{N} \sum_{i \in [N]} X_i^C$. By similar arguments from Eq. (36) to Eq. (38), we have

$$
\sum_{j=1}^{d} (e_k^\mathsf{T} \bar{v}_j^C)^2 \geq \frac{d^{\frac{2}{5}} - \Theta\left(d^{\frac{7}{20}}\right) - O\left(d^{-\frac{1}{5}}\right)}{d^{\frac{2}{5}} - \Theta\left(d^{\frac{7}{20}}\right) + O\left(d^{-\frac{1}{5}}\right)}
$$
$$
= \frac{d^{\frac{2}{5}} - \Theta\left(d^{\frac{7}{20}}\right) + O\left(d^{-\frac{1}{5}}\right)}{d^{\frac{2}{5}} - \Theta\left(d^{\frac{7}{20}}\right) + O\left(d^{-\frac{1}{5}}\right)} - \frac{2O\left(d^{-\frac{1}{5}}\right)}{d^{\frac{2}{5}} - \Theta\left(d^{\frac{7}{20}}\right) + O\left(d^{-\frac{1}{5}}\right)}
$$
$$
= 1 - \frac{O\left(d^{-\frac{1}{5}}\right)}{d^{\frac{2}{5}} - \Theta\left(d^{\frac{7}{20}}\right) + O\left(d^{-\frac{1}{5}}\right)},
\tag{51}
$$

which completes the proof of this lemma.

$\square$

Then, we start to prove Theorem 4.3 as follows.

*Proof.* Lemma A.11 demonstrates the learned local and global representations of distributed CL when positive pairs are generated by similar augmentations. For the other case using dissimilar augmentations, we adopt a similar process to derive the local and global representations.

**For local feature space.** Define $a = x + \xi$ and $b = Hx$. According to the loss function of contrastive learning (CL) with dissimilar augmentations in Section 3.2, we have

$$
\mathcal{L}'_{CL} = -\mathbb{E}_{x \sim D}\left[(W(x + \xi))^\mathsf{T} (WHx)\right] + \frac{1}{2}\|W^\mathsf{T} W\|_F^2
$$
$$
= -\mathbb{E}\left[a^\mathsf{T} W^\mathsf{T} W b\right] + \frac{1}{2}\|W^\mathsf{T} W\|_F^2.
\tag{52}
$$

The minimizer of this loss function is

$$
\frac{\partial \mathcal{L}'_{CL}}{\partial W} = -W\mathbb{E}\left[Hx(x + \xi)^\mathsf{T} + (x + \xi)x^\mathsf{T} H^\mathsf{T}\right] + 2WW^\mathsf{T} W = 0.
\tag{53}
$$

Rearranging it under Assumption A.5 derives

$$
\frac{1}{2}\mathbb{E}\left[Hx(x + \xi)^\mathsf{T} + (x + \xi)x^\mathsf{T} H^\mathsf{T}\right] = W^\mathsf{T} W.
\tag{54}
$$

Let $X_i^{C'}$ denote the left-hand side of the above equation. Hence,

$$
\begin{aligned}
X_i^{C'} &= \frac{1}{2}\mathbb{E}\left[Hx(x+\xi)^\mathsf{T} + (x+\xi)x^\mathsf{T}H^\mathsf{T}\right] \\
&= \frac{1}{2|D_i|}\sum_{j=1}^{|D_i|}\left(Hx_{i,j}x_{i,j}^\mathsf{T} + Hx_{i,j}\xi_{i,j}^\mathsf{T} + x_{i,j}x_{i,j}^\mathsf{T}H^\mathsf{T} + \xi_{i,j}x_{i,j}^\mathsf{T}H^\mathsf{T}\right).
\end{aligned}
\tag{55}
$$

Similarly, based on the formulation that $\xi \sim \mathcal{N}(0, I)$, $\tau = d^{\frac{1}{5}}$ and $\mu = d^{-\frac{1}{5}}$, the expectation of $X_i^{C'}$ can be written as

$$
\mathbb{E}(X_i^{C'}) =
$$
$$
\operatorname{diag}\left(\underbrace{\operatorname{tr}(H)\tau^2 + O\left(d^{-\frac{2}{5}}\right), \dots, \underbrace{\operatorname{tr}(H) + O\left(d^{-\frac{2}{5}}\right)}_{i^{\text{th}}\text{ term}}, \dots, \operatorname{tr}(H)\tau^2 + O\left(d^{-\frac{2}{5}}\right), \dots, O\left(d^{-\frac{2}{5}}\right)}_{N\text{ terms}}\right)
$$
$$
+ \operatorname{diag}\left(\underbrace{O\left(d^{-\frac{1}{5}}\right), \dots, O\left(d^{-\frac{1}{5}}\right)}_{N\text{ terms}}, \dots, O\left(d^{-\frac{1}{5}}\right)\right)
$$
$$
= \operatorname{diag}\left(\operatorname{tr}(H)d^{\frac{2}{5}} + O\left(d^{-\frac{1}{5}}\right), \dots, \operatorname{tr}(H) + O\left(d^{-\frac{1}{5}}\right), \dots, \operatorname{tr}(H)d^{\frac{2}{5}} + O\left(d^{-\frac{1}{5}}\right), \dots, O\left(d^{-\frac{1}{5}}\right)\right).
\tag{56}
$$

Following the proof process from Eqs. (20) to (23), the following inequalities can be found

$$
\begin{aligned}
\left|\lambda_{i,k}^{C'} - \lambda_k\mathbb{E}\left[X_i^{C'}\right]\right| &\le \|X_i^{C'} - \mathbb{E}\left[X_i^{C'}\right]\|_2 \le O\left(d^{-\frac{1}{5}}\right) \\
\operatorname{tr}(H)d^{\frac{2}{5}} - O\left(d^{-\frac{1}{5}}\right) &\le e_k^\mathsf{T}X_i^{C'}e_k \le \operatorname{tr}(H)d^{\frac{2}{5}} + O\left(d^{-\frac{1}{5}}\right) \\
\operatorname{tr}(H)d^{\frac{2}{5}} - O\left(d^{-\frac{1}{5}}\right) &\le \lambda_{i,1}^{C'} \le \operatorname{tr}(H)d^{\frac{2}{5}} + O\left(d^{-\frac{1}{5}}\right).
\end{aligned}
\tag{57}
$$

However, unlike the previous proof, there exists a potential issue that the image transformation matrix $H$ may lead to the case that $X_i^{C'}$ is not a square matrix. Then we denote $X_i^{C'} = \sum_{j=1}^d \lambda_{i,j}u_{i,j}^{C'}v_{i,j}^{C'}$, where $u_{i,j}^{C'}$ and $v_{i,j}^{C'}$ are left and right singular vectors produced by SVD decomposition. So, we have

$$
\begin{aligned}
e_k^\mathsf{T}X_i^{C'}e_k &= \sum_{j=1}^d \lambda_{i,j}(e_k^\mathsf{T}u_{i,j}^{C'}v_{i,j}^{C'}e_k) \\
&\le \lambda_{i,1}^{C'}\sum_{j=1}^d |e_k^\mathsf{T}u_{i,j}^{C'}v_{i,j}^{C'}e_k|,
\end{aligned}
\tag{58}
$$

which further leads to

$$
\begin{aligned}
\sum_{j=1}^d |e_k^\mathsf{T}u_{i,j}^{C'}v_{i,j}^{C'}e_k| &\ge \frac{\operatorname{tr}(H)d^{\frac{2}{5}} - O\left(d^{-\frac{1}{5}}\right)}{\operatorname{tr}(H)d^{\frac{2}{5}} + O\left(d^{-\frac{1}{5}}\right)} \\
&= 1 - \frac{O\left(d^{-\frac{1}{5}}\right)}{\operatorname{tr}(H)d^{\frac{2}{5}} + O\left(d^{-\frac{1}{5}}\right)}.
\end{aligned}
\tag{59}
$$

**For global feature space.** By similar augments from Eq. (27) to Eq. (33) and based on Eq.(59), for the global representation learned through the decentralized learning framework, we establish

$$
\sum_{k=1}^{d} |e_k^\mathsf{T} \bar{u}_j^{C'} \bar{v}_j^{C'} e_k| \geq \frac{\mathrm{tr}(H) \left(1 - \frac{1}{|\bar{A}|}\right) d^{\frac{2}{5}} - O\left(d^{-\frac{1}{5}}\right)}{\mathrm{tr}(H) \left(1 - \frac{1}{|\bar{A}|}\right) d^{\frac{2}{5}} + O\left(d^{-\frac{1}{5}}\right)}
$$
$$
= 1 - \frac{O\left(d^{-\frac{1}{5}}\right)}{\mathrm{tr}(H) \left(1 - \frac{1}{|\bar{A}|}\right) d^{\frac{2}{5}} + O\left(d^{-\frac{1}{5}}\right)}. \tag{60}
$$

On the other hand, for the global objective of the federated learning framework, we follow the arguments from Eq. (36) to Eq. (38) to derive

$$
\sum_{j=1}^{d} |e_k^\mathsf{T} \bar{u}_j^{C'} \bar{v}_j^{C'} e_k| \geq \frac{\mathrm{tr}(H) d^{\frac{2}{5}} - \Theta\left(d^{\frac{7}{20}}\right) - O\left(d^{-\frac{1}{5}}\right)}{\mathrm{tr}(H) d^{\frac{2}{5}} - \Theta\left(d^{\frac{7}{20}}\right) + O\left(d^{-\frac{1}{5}}\right)}
$$
$$
= 1 - \frac{O\left(d^{-\frac{1}{5}}\right)}{\mathrm{tr}(H) d^{\frac{2}{5}} - \Theta\left(d^{\frac{7}{20}}\right) + O\left(d^{-\frac{1}{5}}\right)}. \tag{61}
$$

Combining Lemma A.11, Eq.(59), Eq.(60) and Eq.(61) completes the proof.

$\square$

### A.6.4 PROOF OF FIRST THEORETICAL INSIGHT

This section provides the full proof of Theorem 4.4.

*Proof.* According to Theorem 4.2 and Theorem 4.3, we can find the main difference between the global representations lies in the lower bound. For the global feature learned in the decentralized learning (DecL) framework, we denote the sensitivity of D-SSL as below:

$$
s_{Dec}^{M} = \frac{O\left(d^{-\frac{2}{5}}\right)}{2p\left(1-p\right)\left(1 - \frac{1}{|\bar{A}|}\right) d^{\frac{2}{5}} + O\left(d^{-\frac{2}{5}}\right)}, \tag{62}
$$

$$
s_{Dec}^{C_1} = \frac{O\left(d^{-\frac{1}{5}}\right)}{\left(1 - \frac{1}{|\bar{A}|}\right) d^{\frac{2}{5}} - O\left(d^{-\frac{1}{5}}\right)}, \tag{63}
$$

$$
s_{Dec}^{C_2} = \frac{O\left(d^{-\frac{1}{5}}\right)}{\mathrm{tr}(H)\left(1 - \frac{1}{|\bar{A}|}\right) d^{\frac{2}{5}} - O\left(d^{-\frac{1}{5}}\right)}, \tag{64}
$$

where $s_{Dec}^{M}$ represents the sensitivity of MIM-based D-SSL to heterogeneous data, $s_{Dec}^{C_1}$ represents the sensitivity of CL-based SSL with similar augmentations, and $s_{Dec}^{C_2}$ represents the sensitivity of CL-based SSL with dissimilar augmentations. Then, we compare the magnitude of $s_{Dec}^{M}$ and $s_{Dec}^{C_1}$ by solving the following equation:

$$s_{Dec}^M - s_{Dec}^{C_1} = \frac{O\left(d^{-\frac{2}{5}}\right)}{\left(1 - \frac{1}{|\bar{A}|}\right)d^{\frac{2}{5}} + O\left(d^{-\frac{2}{5}}\right)} - \frac{O\left(d^{-\frac{1}{5}}\right)}{\left(1 - \frac{1}{|\bar{A}|}\right)d^{\frac{2}{5}} - O\left(d^{-\frac{1}{5}}\right)}$$

$$= \frac{O\left(d^{-\frac{2}{5}}\right)\left(\left(1 - \frac{1}{|\bar{A}|}\right)d^{\frac{2}{5}} - O\left(d^{-\frac{1}{5}}\right)\right) - O\left(d^{-\frac{1}{5}}\right)\left(\left(1 - \frac{1}{|\bar{A}|}\right)d^{\frac{2}{5}} + O\left(d^{-\frac{2}{5}}\right)\right)}{\left(\left(1 - \frac{1}{|\bar{A}|}\right)d^{\frac{2}{5}} - O\left(d^{-\frac{1}{5}}\right)\right)\left(\left(1 - \frac{1}{|\bar{A}|}\right)d^{\frac{2}{5}} + O\left(d^{-\frac{2}{5}}\right)\right)}. \tag{65}$$

Consider the dimension $d$ of the Euclidean space is very large so that $d \to \infty$. Then, we have

$$\lim_{d\to\infty}[s_{Dec}^M - s_{Dec}^{C_1}] =$$

$$\lim_{d\to\infty} \frac{O\left(d^{-\frac{2}{5}}\right)\left(\left(1 - \frac{1}{|\bar{A}|}\right)d^{\frac{2}{5}} - O\left(d^{-\frac{1}{5}}\right)\right) - O\left(d^{-\frac{1}{5}}\right)\left(\left(1 - \frac{1}{|\bar{A}|}\right)d^{\frac{2}{5}} + O\left(d^{-\frac{2}{5}}\right)\right)}{\left(\left(1 - \frac{1}{|\bar{A}|}\right)d^{\frac{2}{5}} - O\left(d^{-\frac{1}{5}}\right)\right)\left(\left(1 - \frac{1}{|\bar{A}|}\right)d^{\frac{2}{5}} + O\left(d^{-\frac{2}{5}}\right)\right)} \tag{66}$$

$$= \lim_{d\to\infty} \frac{-\left(1 - \frac{1}{|\bar{A}|}\right)O\left(d^{\frac{1}{5}}\right)}{\left(1 - \frac{1}{|\bar{A}|}\right)^2 \Theta\left(d^{\frac{4}{5}}\right)}.$$

Due to the fact that $2 \leq |\bar{A}| \leq N$, we prove

$$\lim_{d\to\infty}[s_{Dec}^M - s_{Dec}^{C_1}] < 0. \tag{67}$$

Similarly, under Assumption A.9, we determine if $s_{Dec}^M$ is less than $s_{Dec}^{C_2}$ as follows

$$\lim_{d\to\infty}\left[\frac{s_{Dec}^{C_2}}{s_{Dec}^M}\right] = \lim_{d\to\infty} \frac{\frac{O\left(d^{-\frac{1}{5}}\right)}{\text{tr}(H)\left(1 - \frac{1}{|\bar{A}|}\right)d^{\frac{2}{5}} + O\left(d^{-\frac{1}{5}}\right)}}{\frac{O\left(d^{-\frac{2}{5}}\right)}{2p(1-p)\left(1 - \frac{1}{|\bar{A}|}\right)d^{\frac{2}{5}} + O\left(d^{-\frac{2}{5}}\right)}} = \frac{d^{-\frac{3}{5}}}{d^{-\frac{4}{5}}} = \infty, \tag{68}$$

which implies

$$\lim_{d\to\infty}[s_{Dec}^M - s_{Dec}^{C_2}] < 0. \tag{69}$$

Combining Eqs.(67) and (69) arrives

$$\lim_{d\to\infty}[s_{Dec}^M - s_{Dec}^C] < 0, \tag{70}$$

where $s_{Dec}^C$ denotes the sensitivity of CL-based SSL to heterogeneous data. On the other hand, for the federated learning (FL) framework, we denote the following sensitivity of D-SSL:

$$s_{Fed}^M = \frac{O\left(d^{-\frac{2}{5}}\right)}{2p(1-p)d^{\frac{2}{5}} - \Theta\left(d^{\frac{7}{20}}\right) + O\left(d^{-\frac{2}{5}}\right)}, \tag{71}$$

$$s_{Fed}^{C_1} = \frac{O\left(d^{-\frac{1}{5}}\right)}{d^{\frac{2}{5}} - \Theta\left(d^{\frac{7}{20}}\right) + O\left(d^{-\frac{1}{5}}\right)}, \tag{72}$$

$$s_{Fed}^{C_2} = \frac{O\left(d^{-\frac{1}{5}}\right)}{\text{tr}(H)d^{\frac{2}{5}} - \Theta\left(d^{\frac{7}{20}}\right) + O\left(d^{-\frac{1}{5}}\right)}. \tag{73}$$

The difference between $s_{Fed}^M$ and $s_{Fed}^{C_1}$ is given by

$$
\begin{aligned}
s_{Fed}^M - s_{Fed}^{C_1} &= \frac{O\left(d^{-\frac{4}{5}}\right)}{2p\left(1-p\right) - \Theta\left(d^{-\frac{1}{20}}\right) + O\left(d^{-\frac{4}{5}}\right)} - \frac{O\left(d^{-\frac{3}{5}}\right)}{1 - \Theta\left(d^{-\frac{1}{20}}\right) + O\left(d^{-\frac{3}{5}}\right)} \\
&= \frac{O\left(d^{-\frac{4}{5}}\right)\left(1 - \Theta\left(d^{-\frac{1}{20}}\right) + O\left(d^{-\frac{3}{5}}\right)\right) - O\left(d^{-\frac{3}{5}}\right)\left(1 - \Theta\left(d^{-\frac{1}{20}}\right) + O\left(d^{-\frac{4}{5}}\right)\right)}{\left(1 - \Theta\left(d^{-\frac{1}{20}}\right) + O\left(d^{-\frac{4}{5}}\right)\right)\left(1 - \Theta\left(d^{-\frac{1}{20}}\right) + O\left(d^{-\frac{3}{5}}\right)\right)} \\
&= \frac{-O\left(d^{-\frac{3}{5}}\right) + \Theta\left(d^{-\frac{13}{20}}\right)}{d^{\frac{1}{5}} - \Theta\left(d^{\frac{3}{20}}\right) + O\left(d^{-\frac{3}{5}}\right)}.
\end{aligned}
\tag{74}
$$

For the above result, let $d \to \infty$, we can establish

$$\lim_{d\to\infty}[s_{Fed}^M - s_{Fed}^{C_1}] = \lim_{d\to\infty} \frac{-O\left(d^{-\frac{3}{5}}\right) + \Theta\left(d^{-\frac{13}{20}}\right)}{d^{\frac{1}{5}} - \Theta\left(d^{\frac{3}{20}}\right) + O\left(d^{-\frac{3}{5}}\right)} = \lim_{d\to\infty} \frac{-O\left(d^{-\frac{3}{5}}\right)}{d^{\frac{1}{5}}} < 0 \tag{75}$$

Then for the comparison between $s_{Fed}^M$ and $s_{Fed}^{C_2}$, under Assumption A.9, we have

$$\lim_{d\to\infty}[\frac{s_{Fed}^{C_2}}{s_{Fed}^M}] = \lim_{d\to\infty} \frac{\dfrac{O\left(d^{-\frac{1}{5}}\right)}{\text{tr}(H)\,d^{\frac{2}{5}} - \Theta\left(d^{\frac{7}{20}}\right) + O\left(d^{-\frac{1}{5}}\right)}}{\dfrac{O\left(d^{-\frac{1}{5}}\right)}{d^{\frac{2}{5}} - \Theta\left(d^{\frac{7}{20}}\right) + O\left(d^{-\frac{1}{5}}\right)}} = \frac{d^{-\frac{3}{5}}}{d^{-\frac{4}{5}}} = \infty. \tag{76}$$

With Eqs.(75) and (76), we find

$$\lim_{d\to\infty}[s_{Fed}^M - s_{Fed}^C] < 0. \tag{77}$$

Combining Eq.(70) and Eq.(77) completes the proof.

$\square$

### A.6.5 PROOF OF SECOND THEORETICAL INSIGHT

This section provides the full proof of Corollary 4.5 and Theorem 4.6.

*Proof.* For the decentralized learning (DecL) framework, we notice from Eqs.(62), (63) and (64) that their denominators both include the term $1 - \frac{1}{|\bar{A}|}$. Since $|\bar{A}|$ is proportional to $1 - \frac{1}{|\bar{A}|}$, we derive that $|\bar{A}|$ is inversely proportional to $s_{Dec}^M$, $s_{Dec}^{C_1}$ and $s_{Dec}^{C_2}$, which completes the proof of Corollary 4.5. Next, by a similar proof from Eq.(62) to Eq.(77), we compare the robustness of distributed MIM between DecL and FL framework by solving

$$s_{Dec}^M - s_{Fed}^M = \frac{O\left(d^{-\frac{2}{5}}\right)}{2p\left(1-p\right)\left(1 - \frac{1}{|\bar{A}|}\right)d^{\frac{2}{5}} + O\left(d^{-\frac{2}{5}}\right)} - \frac{O\left(d^{-\frac{2}{5}}\right)}{2p\left(1-p\right)d^{\frac{2}{5}} - \Theta\left(d^{\frac{7}{20}}\right) + O\left(d^{-\frac{2}{5}}\right)}. \tag{78}$$

This is equivalent to solving

$$2p\left(1-p\right)\left(1 - \frac{1}{|\bar{A}|}\right)d^{\frac{2}{5}} - \left(2p\left(1-p\right)d^{\frac{2}{5}} - \Theta\left(d^{\frac{7}{20}}\right)\right)$$
$$= 2p\left(1-p\right)d^{\frac{2}{5}} - \frac{2p\left(1-p\right)}{|\bar{A}|}d^{\frac{2}{5}} - 2p\left(1-p\right)d^{\frac{2}{5}} + \Theta\left(d^{\frac{7}{20}}\right). \tag{79}$$

Due to the fact that

$$\lim_{d\to\infty}\left[2p\left(1-p\right)d^{\frac{2}{5}} - \frac{2p\left(1-p\right)}{|\bar{A}|}d^{\frac{2}{5}} - 2p\left(1-p\right)d^{\frac{2}{5}} + \Theta\left(d^{\frac{7}{20}}\right)\right] < 0, \tag{80}$$

we have

$$\lim_{d\to\infty}\left[s_{Dec}^M - s_{Fed}^M\right] > 0. \tag{81}$$

Similarly, for CL-based SSL, we have

$$s_{Dec}^{C_1} - s_{Fed}^{C_1} = \frac{O\left(d^{-\frac{1}{5}}\right)}{\left(1 - \frac{1}{|\bar{A}|}\right)d^{\frac{2}{5}} - O\left(d^{-\frac{1}{5}}\right)} - \frac{O\left(d^{-\frac{1}{5}}\right)}{d^{\frac{2}{5}} - \Theta\left(d^{\frac{7}{20}}\right) + O\left(d^{-\frac{1}{5}}\right)}, \tag{82}$$

$$s_{Dec}^{C_2} - s_{Fed}^{C_2} = \frac{O\left(d^{-\frac{1}{5}}\right)}{\text{tr}(H)\left(1 - \frac{1}{|\bar{A}|}\right)d^{\frac{2}{5}} - O\left(d^{-\frac{1}{5}}\right)} - \frac{O\left(d^{-\frac{1}{5}}\right)}{\text{tr}(H)d^{\frac{2}{5}} - \Theta\left(d^{\frac{7}{20}}\right) + O\left(d^{-\frac{1}{5}}\right)}. \tag{83}$$

Under Assumption A.9, the above results imply that

$$\lim_{d\to\infty}\left[s_{Dec}^{C_1} - s_{Fed}^{C_1}\right] > 0, \tag{84}$$

$$\lim_{d\to\infty}\left[s_{Dec}^{C_2} - s_{Fed}^{C_2}\right] > 0. \tag{85}$$

With Eqs.(84) and (85), we find

$$\lim_{d\to\infty}\left[s_{Dec}^C - s_{Fed}^C\right] > 0. \tag{86}$$

Combining Eq.(81) with Eq.(86) derives

$$\lim_{d\to\infty} [s_{Dec} > s_{Fed}]. \tag{87}$$

Note that Eq.(87) holds for decentralized learning setups in which each client has an inconsistent number of neighbors. However, there exists an optimal case for decentralized learning, denoted by $\forall i, |A_i| = N$. In this case, the global objective of decentralized learning can be re-formulated as follows:

$$\sum_{i\in[N]} \frac{1}{N} \sum_{j\in[N]} \frac{1}{N}\mathcal{L} = \sum_{i\in[N]} \frac{1}{N}\mathcal{L}. \tag{88}$$

This equation is exactly the same as the global objective of federated learning shown in Eq.(1). Therefore, we know the below statement holds:

$$\lim_{d\to\infty} [s_{Dec} = s_{Fed}], \tag{89}$$

when $\forall i \in [N], |A_i| = N$. Combining Eq.(87) and Eq.(89) completes the proof.

$\square$

