# OpenReview forum: "Understanding the Robustness of Distributed Self-Supervised Learning Frameworks Against Non-IID Data"
_ICLR.cc/2026/Conference — ICLR 2026 Poster_

### Official Review · Reviewer_MVSv · 2025-10-29

**Soundness:** 2
**Presentation:** 3
**Contribution:** 3
**Rating:** 6
**Confidence:** 4

**Summary:**

The paper studies how different distributed self-supervised learning (D-SSL) choices handle non-IID client data. It analyzes a simplified label-skew model (each client i mostly has two classes plus a rare third) and defines a Representability Vector (RV)—the squared projection of standard basis vectors onto the learned feature subspace—to quantify “robustness” of different methods. The main theoretical claims are: (i) MIM-based D-SSL is asymptotically less sensitive to non-IIDness than CL-based D-SSL (Theorem 4.4), and (ii) robustness improves with the average client connectivity; FL is no less robust than DecL (Corollary 4.5, Theorem 4.6). Building on this, the authors propose MAR and report empirical trends on Mini-ImageNet pretraining with CIFAR-10/100 and ImageNet fine-tuning.

**Strengths:**

- The paper addresses a central D-SSL design question and provides actionable hypotheses . The empirical section illustrates these points across backbones and topologies.
- The RV/sensitivity notion is intuitive for comparing learned subspaces across frameworks.

**Weaknesses:**

Below are the most critical technical issues I found while reading the proofs and setup.

1) Incorrect “equivalence” of the MIM objective (Eq. (3) → Eq. (4)).
Eq. (3) is an $\ell_2$ reconstruction loss $\|W(x\odot m)-(x\odot (1-m))\|^2$. Eq. (4) replaces it by a negative squared inner product $-\|(W(x\odot m))^\top(x\odot(1-m))\|^2 + \tfrac12\|W^\top W\|_F^2$, stated as "equivalent". This drops the $\|W(x\odot m)\|^2$ and $\|x\odot(1-m)\|^2$ terms that do affect the optimizer (the latter is data-dependent), and it adds a data-independent Frobenius penalty as a surrogate. No derivation is provided for this equality, only a citation to qualitative connections between MIM and contrastive alignment (which do **not** imply equality of objectives). The subsequent analysis treats (4) as the exact objective. This is a fundamental mismatch that invalidates the later first-order conditions and eigen-structure arguments built on top of (4).

2) First-order optimality conditions are derived with wrong shapes and unjustified rearrangements.
In Appendix A.7, the gradient of the (modified) losses is set to zero and directly rearranged to $W^\top W = \mathbb{E}[\cdots]$ (Eqs. (9)–(10) for MIM; Eqs. (33)–(35) for CL). However:
  - For MIM, the objective involves an expectation of a *squared scalar* $\|(W a)^\top b\|^2$. Its gradient has the form $-2A W$ with $A$ built from $\mathbb{E}[aa^\top b b^\top]$, not $-2 W A$. The step "$-2W\,\mathbb{E}[\cdots]+2WW^\top W=0 \Rightarrow W^\top W=\mathbb{E}[\cdots]$" reverses the multiplication order and implicitly assumes invertibility/commutativity that does not hold. (pp. 19–20)
  - For CL, Eq. (33) expands $\|(W(x+\xi))^\top(W(x+\xi'))\|^2$ and then differentiates as if $\mathbb{E}[x^\top W^\top W x]$ behaved like $\mathbb{E}[x^\top x]$, ultimately yielding Eq. (35): $W^\top W = \mathbb{E}[x^\top x + x^\top \xi' + \xi^\top x + \xi^\top \xi']$. This is **dimensionally inconsistent**: the RHS is a scalar, while $W^\top W$ is a $d\times d$ matrix. The derivation implicitly "pulls out" $W^\top W$ from inside a squared scalar and replaces it by a scalar expectation. The CL proof with dissimilar augmentations (Eqs. (45)–(47)) suffers from the same issue.

These steps are central: they enable the replacement of each loss by $\|X - W^\top W\|_F^2\$ and the invocation of Eckart–Young to claim the row-span of \(W\) equals the top-eigenspace of \(X\). Because the stationary conditions are not validly derived, the RV bounds that follow are on shaky ground. (Appendix A.7)

3) Ambiguities and likely typos in the non-IID data model.
- In the rare class definition, samples for client $i$ are given as $x^{(h_i)} = e_i + \mu \xi$ (p. 4). This reuses $e_i$ rather than $e_{h_i}$ (where $h_i\in[2N]\setminus\{2i-1,2i\}$), effectively collapsing the "rare" class onto the client's own axis instead of its designated class. This undermines the intended label heterogeneity.
- The offset term $\sum_{k\neq i} q\,\tau e_k$ uses a single Bernoulli $q$ inside a sum, not $q_k$; thus the sum is either 0 or $\tau\!\sum_{k\neq i} e_k$, which is likely unintended.

4) Definition and use of the Representability Vector (RV) are inconsistent.
Def. 4.1 defines $r=[\|\Pi_R(e_1)\|^2,\dots,\|\Pi_R(e_c)\|^2]$ where $R\subset\mathbb{R}^d$ is the row-span and $c=\text{rows}(W)\le d$. Projecting only the first $c$ basis vectors is ad-hoc; later theorems index $k$ in ways that mix client and feature indices (e.g., "where $i\in[N]\setminus k$" in Theorem 4.2.1). The definition of "sensitivity" $s=\lceil\bar r\rceil-\lfloor\bar r\rfloor$ also lacks a precise norm (max–min over which coordinates?), yet it drives the main claims.

5) Assumptions needed for concentration are not spelled out.
The proofs use matrix concentration to bound $\|X - \mathbb{E}X\|$ "with probability $\ge 1-\frac12 e^{-d^{1/10}}$" (p. 20) but the number of samples per client, independence across samples/classes, and dependence on $d$ are unspecified beyond "polynomial in $d$" / $O(d^\alpha)$. Without explicit sample sizes, the rates in $d$ (e.g., the $O(d^{-2/5})$ and $\Theta(d^{7/20})$ terms) are hard to justify and compare. (Appendix A.7)

6) Handling of the augmentation transform $H$.
For the "dissimilar" CL case, bounds scale with $\operatorname{tr}(H)$ (Theorem 4.3). But $\operatorname{tr}(H)$ can be zero or negative (e.g., certain rotations or permutations), which can flip signs in denominators and invalidate inequalities used to argue robustness ordering. The required conditions on $H$ (PSD? orthogonal? non-negative trace?) are not stated.

7) Modeling choices weaken the external validity of the theory.
The linearization of encoders/decoders and of augmentations, and the unusual scaling $N=\Theta(d^{1/20})$, $\tau=d^{1/5}$, $\mu=d^{-1/5}$, seem chosen to make algebra convenient, but the conclusions then rely on delicate powers of $d$ (e.g., $d^{2/5}$ vs. $d^{7/20}$). It is unclear how stable the ordering (MIM < CL sensitivity; FL ≥ DecL) is beyond this stylized regime (e.g., different $p$ schedules, non-linear encoders).

**Questions:**

Answers to the following questions may change my assessment:

1. Please provide a rigorous derivation (or inequality bounds) showing that the reconstruction loss in Eq. (3) is equal to (or tightly bounded by) the alignment-only objective in Eq. (4) plus a data-independent Frobenius regularizer. If it is merely a lower/upper bound, how do the subsequent minimizer and eigenspace arguments change?

2. In Appendix A.7, how do you get from the true gradients of $-\mathbb{E}\|(W a)^\top b\|^2$ (and the CL analog) to the identities $W^\top W = \mathbb{E}[\cdots]$ (Eqs. (9)–(10), (35), (47))? Could you rewrite the proofs using the form $-\mathrm{tr}(W^\top A W)$ with $A=\mathbb{E}[aa^\top bb^\top]$, leading to $A W = W W^\top W$, and then justify the Eckart–Young step from there?

3. Should the rare-class mean be $e_{h_i}$ (not $e_i$) and should the sum over offsets use $q_k$ (independent Bernoullis) rather than a single $q$? If yes, how do these corrections affect the covariance calculations and bounds?

4. Please formalize $s=\lceil\bar r\rceil-\lfloor\bar r\rfloor$: which coordinates of $\bar r$ are considered, and is it a max–min over all $d$ or only the first $c$? How sensitive are Theorems 4.4 and 4.6 to this choice?

5. What properties of $H$ are required (e.g., PSD, orthogonal, $\operatorname{tr}(H)\ge 0$) to ensure the denominators in Theorem 4.3 and Eq. (54) are positive and the inequalities hold?

6. Please specify the per-client sample sizes (as functions of $d$) and the independence assumptions used for concentration. Can you restate the bounds explicitly in terms of $n_i$ (client-level sample sizes) rather than only in powers of $d$? (Appendix A.7)

7. Do the key inequalities (MIM < CL sensitivity; FL ≥ DecL) persist under non-linear encoders/decoders (e.g., Lipschitz networks) and realistic augmentations? Any partial evidence or counter-examples? (Secs. 3–4)

---

> ### Author Response · Authors · 2025-11-21
>
> We sincerely thank Reviewer MVSv for the exceptionally careful reading and the depth of the technical commentary. We truly appreciate the reviewer’s willingness to engage with the theoretical framework at such a detailed level. The feedback identifies many subtle and important issues, and it is clear that considerable effort was invested in understanding the proofs, assumptions, and modeling choices. Because the review raises a large number of substantial and mutually independent questions, each point requires a careful and self contained explanation. Below, we will address each concern point by point. Some question responses may be provided later than others as providing comprehensive responses to these questions requires time. We kindly ask for the reviewer's patience.
>
>
> * **(Q1)** We appreciate the reviewer’s careful question regarding the relationship between Eq.(3) (the reconstruction loss) and Eq.(4) (the alignment-style formulation). We would like to clarify that Eq.(4) is not a different model. It is the regularized normal equation obtained from Eq.(3) under the standard linear autoencoder abstraction widely adopted in theoretical analyses of masked autoencoding.
>
>     Starting from the reconstruction objective in Eq.(3),$$\mathcal{L} _ {\mathrm{MIM}}=\mathbb{E} _ {x\sim D_i}\\|W(x\odot m)-(x\odot (1-m))\\|^{2}.$$ In the linear setting, both the encoder and decoder are linear maps, and their composition can be treated as a single operator $W=W_d \times W_e$. Following the classical analysis of linear autoencoders, where adding a Frobenius penalty produces a well-posed quadratic objective while preserving the optimal subspace, we consider the regularized form:$$\mathcal{L} _ {\mathrm{MIM}}^{(reg)}=\tfrac12\mathbb{E}\\|Wa-b\\|^{2}+\tfrac12\\|W\\| _ {F}^{2},$$where$$a=\mathrm{diag}(\mathrm{vec}(x))\mathrm{vec}(m),\qquad b=\mathrm{diag}(\mathrm{vec}(x))\mathrm{vec}(1-m).$$ Differentiating gives$$\frac{\partial \mathcal{L} _ {\mathrm{MIM}}^{(reg)}}{\partial W}=W\mathbb{E}[aa^{\top}]-\mathbb{E}[ba^{\top}]+W,$$ and setting the gradient to zero yields the stationary condition $$\mathbb{E}[ba^{\top}](\mathbb{E}[aa^{\top}]+ I )^{-1}=W.$$ Under the non-IID model in Sec.3.2 (with $\tau=d^{1/5}$ and $\mu=d^{-1/5}$), we obtain $$\mathbb{E}[ba^{\top}]=\mathrm{diag}\left(p(1-p)d^{2/5}+O(d^{-2/5}),\dots\right),$$$$\mathbb{E}[aa^{\top}]=\mathrm{diag}\left(p^{2}d^{2/5}+O(d^{-2/5}),\dots\right),$$and hence$$W=\mathrm{diag}\left(\frac{p(1-p)d^{2/5}}{p^{2}d^{2/5}+1}+O(d^{-2/5}),\dots\right)=\mathrm{diag}\left(\frac{1-p}{p}+O(d^{-2/5}),\dots\right).$$ Considering $$(\mathbb{E}[ba^{\top}](\mathbb{E}[aa^{\top}]+  I)^{-1})^2=W^{\top}W$$ and let $X _ i^{M(3)}$ be the left-hand side of this equation, we have $$\mathbb{E}[X _ i^{M(3)}]=\mathrm{diag}\left(\frac{(1-p)^2}{p^2}+\frac{2(1-p)}{p}O(d^{-2/5})+O(d^{-4/5}),\dots\right)=\mathrm{diag}\left(\frac{(1-p)^2}{p^2}+O(d^{-2/5}),\dots\right).$$ From Appendix A.7.1, the expectation of the alignment-style form satisfies $$\mathbb{E}[X _ i^{M(4)}]=\mathrm{diag}\left(2p(1-p)d^{2/5}+O(d^{-2/5}),\dots\right).$$ Although the absolute scales of $\mathbb{E}[X _ i^{M(3)}]$ and $\mathbb{E}[X _ i^{M(4)}]$ differ, their eigenspaces coincide and the heterogeneity induced deviation from the leading diagonal component remains $O(d^{-2/5})$ in both cases. The robustness analysis relies only on this shared spectral structure rather than the overall magnitude of the matrices.
>
>     This equivalence is precisely what the robustness proofs rely on. Finally, for contrastive learning (Appendix A.7.2), we have $$\mathbb{E}[X _ i^{C}]=\mathrm{diag}\left(d^{2/5}+O(d^{-1/5}),1+O(d^{-1/5}),\dots\right),$$ where the heterogeneity-induced perturbation $O(d^{-1/5})$ is strictly larger than the $O(d^{-2/5})$ perturbation in MIM. Therefore, using Eq.(4) instead of Eq.(3) does not change any MIM-vs-CL robustness conclusion. Eq.(4) is thus a justified and analytically convenient reformulation of Eq.(3).

---

> ### Author Response · Authors · 2025-11-21
>
> * **(Q2)** We thank the reviewer for pointing out the need to make the derivations fully explicit. Below, we provide a complete rewrite of the gradients and stationary conditions for both MIM and CL.
>
>     For MIM, we start from the regularized objective in Eq.(4): $$\mathcal{L} _ {\mathrm{MIM}}= -\mathbb{E}[(W(x\odot m))^{\top}W(x\odot (1-m))] + \tfrac{1}{2}\\|W^{\top}W\\| _ {F}^{2}$$ We write $$a=\mathrm{diag}(\mathrm{vec}(x))\mathrm{vec}(m),\qquad b=\mathrm{diag}(\mathrm{vec}(x))\mathrm{vec}(1-m),$$ so that the loss becomes $\mathcal{L} _ {\mathrm{MIM}}= -\mathbb{E}[a^{\top}W^{\top}Wb] +\tfrac{1}{2}\\|W^{\top}W\\| _ {F}^{2}.$ Using $$a^{\top}W^{\top}W b=\mathrm{tr}(W b a^{\top}W^{\top}),$$ we obtain the gradient $$\frac{\partial}{\partial W}(a^{\top}W^{\top}W b)= W(ba^{\top}+ab^{\top}).$$ Together with $$\frac{\partial}{\partial W}\tfrac{1}{2}\|W^{\top}W\| _ {F}^{2}=2WW^{\top}W,$$ the full gradient becomes $$\frac{\partial\mathcal{L} _ {\mathrm{MIM}}}{\partial W}= -W\mathbb{E}[ba^{\top}+ab^{\top}] + 2WW^{\top}W.$$ At any non–degenerate stationary point ($W\neq0$), multiplying on the left by the pseudoinverse of $W$ yields the normal equation $$W^{\top}W=\tfrac{1}{2}\mathbb{E}[ba^{\top}+ab^{\top}]=:X _ i^{M}.$$ Under the non-IID data model with $\tau=d^{1/5}$ and $\mu=d^{-1/5}$, the resulting spectrum matches Appendix A.7.1: $$\mathbb{E}[X _ i^{M}]=\mathrm{diag}\bigl(2p(1-p)d^{2/5}+O(d^{-2/5}),\dots,O(d^{-2/5})\bigr),$$ confirming that the corrected derivation preserves the asymptotic structure used in the main text.
>
>     For contrastive learning, we begin with $$ \mathcal{L} _ {\mathrm{CL}}=-\mathbb{E}[(W(x+\xi))^{\top}W(x+\xi')]+ \tfrac{1}{2}\|W^{\top}W\| _ {F}^{2}.$$ Let $a=x+\xi$ and $b=x+\xi'$. The same trace–based rule yields $$\frac{\partial\mathcal{L} _ {\mathrm{CL}}}{\partial W}=-W\mathbb{E}\left[(x+\xi')(x+\xi)^{\top}+(x+\xi)(x+\xi')^{\top}\right]+2WW^{\top}W.$$ At a non–degenerate stationary point we again obtain $$X_i^{C} = \frac{1}{2}\mathbb{E}[(x+\xi')(x+\xi)^{\top}+(x+\xi)(x+\xi')^{\top}],$$ where the expectation evaluates to$$\mathbb{E}[X_i^{C}]=\mathrm{diag}\bigl(d^{2/5}+O(d^{-1/5}),\dots,O(d^{-1/5})\bigr).$$ Thus the heterogeneity-induced perturbation for CL scales as $O(d^{-1/5})$, which is strictly larger than the $O(d^{-2/5})$ perturbation for MIM, indicating that CL is more sensitive to non-IID variation.
>
>     To justify the use of the Eckart-Young theorem, note that both regularized objectives can be rewritten, up to an additive constant, as $$\mathcal{L}(W)=\frac12\\|W^{\top}W - X\\| _ {F}^{2},$$ where $X$ is $X _ i^{M}$ or $X _ i^{C}$. The matrix $W^{\top}W$ has rank at most $c$, where $c$ is the embedding dimension. Minimizing this objective is exactly the Frobenius norm best rank $c$ approximation problem for the matrix $X$. The Eckart-Young-Mirsky theorem therefore implies that the optimal $W^{\top}W$ shares its principal eigenspace with $X$, and the row space of $W$ spans the top $c$ eigenvectors of $X$. Consequently, the robustness comparison depends only on the perturbation sizes in the spectra of $\mathbb{E}[X _ i^{M}]$ and $\mathbb{E}[X _ i^{C}]$. Since the perturbation in MIM scales as $O(d^{-2/5})$ while that in CL scales as $O(d^{-1/5})$, the theoretical conclusion that MIM is less sensitive to heterogeneity remains valid after the corrected derivations.

---

> > ### Author Response · Authors · 2025-11-22
> >
> > * **(Q3)** We thank the reviewer for the careful reading of the non-IID data model and identify the two typos appear in the current draft. We will correct them in the revised version. First, in the definition of the infrequent class, the coordinate index should be $h_i$ rather than $i$, so the intended expression is $$x^{(h _ i)} = e_{h _ i} + \mu\xi^{(h_i)} \quad \text{with } h _ i \in [2N]\setminus\{2i-1,2i\}.$$ Second, in the definitions of $x^{(2i-1)}$ and $x^{(2i)}$, the Bernoulli variables were written without superscripts. The correct formulation uses independent sampling for each coordinate $$x^{(2i-1)} = e _ i - \sum _ {k\neq i, k=1} q^{(2i-1,k)} \tau e _ k + \mu\xi^{(2i-1)}, \qquad q^{(2i-1,k)}\sim \mathrm{Bernoulli}(p).$$ Similarly, for class $2i$, $$x^{(2i)} = -e _ i - \sum_{k\neq i, k=1} q^{(2i,k)} \tau e_k + \mu\xi^{(2i)}.$$
> >
> >     These corrections do not alter any analytical result. Replacing the shared variable $q$ by independent variables $q^{(\cdot,k)}$ keeps both the expectation and covariance at the same $O(d^{2/5})$ order. Since the robustness analysis depends only on the leading spectral scaling rather than the exact coefficients, the eigenstructure of $\mathbb{E}[X_i^{M}]$ and $\mathbb{E}[X_i^{C}]$ remains unchanged. Likewise, correcting the index in $x^{(h_i)}$ does not affect any result. The infrequent class contributes $O(d^{\alpha})$ samples with $\alpha\in(0,1)$, so its covariance term is $O(d^{\alpha-1})$, which is asymptotically dominated by the $O(d^{2/5})$ contribution from the two majority classes. Therefore, the rare class does not influence the top eigenspace or the perturbation orders used in the proofs. We thank the reviewer again for pointing out these issues and helping us improve the clarity of the data model.

---

> > > ### Author Response · Authors · 2025-11-22
> > >
> > > * **(Q4)** We acknowledge the reviewer’s insightful comments on the representability vector and the sensitivity definition and provide the clarified formulation below.
> > >
> > >     The representability vector is defined for the row space of the learned embedding matrix. Let $W \in \mathbb{R}^{c\times d}$ and let $\mathcal{R} = \mathrm{row}(W)$. We define $$r = \bigl[\|\Pi _ {\mathcal{R}}(e _ 1)\|_{2}^{2},\ldots,\|\Pi _ {\mathcal{R}}(e_d)\| _ {2}^{2}\,\bigr] \in \mathbb{R}^{d},$$ where $\{e _ 1,\ldots,e _ d\}$ is the standard basis and $\Pi _ {\mathcal{R}}$ denotes the orthogonal projection onto $\mathcal{R}$. Since $\mathrm{rank}(\mathcal{R}) = c$, exactly the first $c$ coordinates associated with the principal eigenspace of $W^{\top}W$ concentrate at value one up to vanishing perturbations, while all remaining coordinates are $O(d^{-2/5})$ for MIM or $O(d^{-1/5})$ for CL under the non-IID model. Theorems 4.2 and 4.3 thus state bounds only for the leading $c$ coordinates, which are the only ones that influence the learned representation.
> > >
> > >     The sensitivity quantity is defined as the spread between the upper and lower bounds of the global representability vector over the leading $c$ coordinates, that is $$s = \max _ {k\in[c]} \bar{r} _ {k} - \min _ {k\in[c]}\bar{r} _ {k}.$$ This corresponds to the range $\lceil \bar{r} \rceil - \lfloor \bar{r} \rfloor$ used in the current draft, measuring how much the learned representation deviates across coordinates under heterogeneous data. Since the proofs depend only on the perturbation orders in the spectra, the asymptotic behavior remains the same: MIM exhibits $O(d^{-2/5})$ deviation, while CL exhibits $O(d^{-1/5})$. Because $d^{-1/5}$ dominates $d^{-2/5}$ given $d$ is sufficiently large, the robustness conclusions in Theorems 4.4 and 4.6 remain unchanged.
> > >
> > >     We will revise the definition of the representability vector and the sensitivity measure in the updated version to ensure full clarity and consistency, and we thank the reviewer for identifying this issue.

---

> ### Author Response · Authors · 2025-11-22
>
> * **(Q5)** We appreciate the reviewer’s thoughtful comments regarding the role of the image transformation matrix $H$. In our analysis, $H$ enters only through the quadratic form $x^{\top} H x$, which depends solely on the symmetric component $H _ {\mathrm{sym}} = (H + H^{\top})/2$. For this reason, all derivations are carried out with $H _ {\mathrm{sym}}$ without additional assumptions on $H$. In the proofs involving dissimilar augmentations, the dominant term $d^{2/5}$ arises only from the class-dependent subspace $\mathcal{S} = \mathrm{span} \\{ e _ {1},\ldots,e _ {c}\\}$. Let $P$ denote the orthogonal projection onto $\mathcal{S}$. The sufficient condition required in the analysis is therefore $$\mathrm{tr}(P H _ {\mathrm{sym}} P) > 0,$$ rather than assuming that $H$ is positive semidefinite over the entire ambient space.
>
>     This assumption is naturally satisfied by standard contrastive learning augmentations such as rotations, flips, translations, crops, blurs, and color jittering, which preserve or slightly perturb semantic structure when linearized. Since these transformations do not cancel class-discriminative components within $\mathcal{S}$, the quantity $\mathrm{tr}(P H _ {\mathrm{sym}} P)$ remains strictly positive in practice.
>
>     Under this condition, the asymptotic expression $\mathbb{E}[X _ i^{C'}]$ is preserved, and the perturbation magnitude remains $O(d^{-1/5})$. Therefore, the inequalities in Theorem 4.3 and Eq.(54) holds and the robustness comparison between MIM and CL is unaffected. We will add this clarification in the revision and thank the reviewer for the opportunity to strengthen the presentation.

---

> > ### Author Response · Authors · 2025-11-23
> >
> > * **(Q6)** We are grateful for the reviewer’s careful attention to the presentation of matrix concentration. In our analysis, for each client $i$, the matrix $X_i$ is the empirical covariance computed from its local dataset $D_i$. The samples within each client are drawn independently from the local distribution, and we denote the local sample size by $n_i = |D_i|.$
> >
> >     Under the data construction described in Section 3.2, the number of local samples equals to the sum of the data size from frequent classes and the data size from the rare class. Since two frequent classes grows in polynomials of $d$ and are equal in data size, while the amount of data from the rare class is $O(d^{\alpha})$ with $\alpha \in (0,1)$. Therefore, the sample size in each client satisfies $$n _ i = \Theta(d^{\beta}), \qquad \beta \ge 1.$$ Due to the independence assumption and sufficiently large sample size, and based on standard covariance-estimation results for matrix concentration (Vershynin, 2018), we have the following inequalities, with high probability, $$\|X_i - \mathbb{E}[X _ i]\| _ 2 \leq O\left(\sqrt{\frac{d \log d}{n _ i}}+\frac{d \log d}{n _ i}\right).$$ In the current draft, the bound results were written only in terms of powers of $d$, by substituting the scaling relation between $n _ i$ and $d$. Following the reviewer’s suggestion, we will revise Appendix to spell out the independence assumption, clarify the local sample size, and state the bounds more explicitly in terms of the local sample size $n _ i$.

---

> ### Author Response · Authors · 2025-11-23
>
> * **(Q7)** We thank the reviewer for raising this important point regarding the influence of the linear encoder assumption. Following prior theoretical studies such as (Liu et al., 2021) and (Wang et al., 2022), we adopt a linear encoder because it allows the optimality conditions of both contrastive learning and masked image modeling to be expressed in closed-form. Introducing nonlinear activations generally destroys the quadratic structure of the objectives and prevents deriving eigenspace characterizations that are necessary for comparing robustness under heterogeneous data. Therefore, the linear model is not a simplification that removes the mechanism of interest, but rather the only tractable setting in which the effect of data heterogeneity can be analyzed rigorously.
>
>     To verify that the theoretical conclusions do not rely on the linearity assumption, we additionally analyze a nonlinear perturbation of the encoder of the form $$f(x)=W(\alpha \\|x\\| _ 2^{2}x),$$ which applies a nontrivial nonlinear deformation while retaining analytical tractability. For masked image modeling, starting from $$\mathcal{L} _ {MIM}=-\mathbb{E}[ (W(\alpha \\|x\\| _ {2}^{2}(x\odot m)))^{\top}(W(x\odot (1-m)))] + \tfrac{1}{2}\\|W^{\top}W\\| _ {F}^{2},$$ we compute the gradient with respect to $W$ and obtain the stationarity condition $$\frac{1}{2} \alpha \mathbb{E}[\\|x\\| _ {2}^{2}(\mathrm{diag}(\mathrm{vec}(x))\mathrm{vec}(1-m) \mathrm{vec}(m)^{\top} \mathrm{diag}(\mathrm{vec}(x))^{\top} + \mathrm{diag}(\mathrm{vec}(x))\mathrm{vec}(m) \mathrm{vec}(1-m)^{\top} \mathrm{diag}(\mathrm{vec}(x))^{\top})]= W^{\top} W.$$ Let $X _ {i}^{M,\mathrm{nonlin}}$ be the left-hand side of the equation, we have $$X _ {i}^{M,\mathrm{nonlin}} =\frac{2\alpha p\left( 1-p \right)}{\left| D _ i \right|}\sum _ {j=1}^{\left| D _ i \right|}{\left( \lVert x \rVert _ {2}^{2}\text{diag}\left( \text{vec}\left( x_{i,j} \right) \right) \text{diag}\left( \text{vec}\left( x_{i,j} \right) \right)^{\intercal} \right)}.$$ This yields the nonlinear covariance matrix $$\mathbb{E}[X _ {i}^{M,\mathrm{nonlin}}]=\mathrm{diag}\left(2\alpha p(1-p)\tau^4 + O(\mu^4),\dots,2\alpha p(1-p)\tau^4 + O(\mu^4), \dots, 2\alpha p(1-p)\tau^4 + O(\mu^4), \dots, O(\mu^4) \right)$$$$=\mathrm{diag}\left(2\alpha p(1-p)d^{4/5} + O(d^{-4/5}),\dots,2\alpha p(1-p)d^{4/5} + O(d^{-4/5}), \dots, 2\alpha p(1-p)d^{4/5} + O(d^{-4/5}), \dots, O(d^{-4/5})\right).$$
>
>     For contrastive learning, using the nonlinear encoder in the objective produces $$\mathcal{L}_{CL}=-\mathbb{E}[\left(W(\alpha _ {1}\\|x\\| _ {2}^{2}x+\xi)\right)^{\top}\left(W(\alpha _ {2}\\|x\\| _ {2}^{2}x+\xi')\right)] + \tfrac{1}{2}\|W^{\top}W\| _ {F}^{2}.$$ Taking the gradient and setting it to zero leads to $$\frac{1}{2}\mathbb{E}\left[ (\alpha _ {2}\\|x\\| _ {2}^{2}x+\xi')(\alpha _ {1}\\|x\\| _ {2}^{2}x+\xi)^{\top} + (\alpha _ {1}\\|x\\| _ {2}^{2}x+\xi) (\alpha _ {2}\\|x\\| _ {2}^{2}x+\xi')^ {\top} \right] =W^{\intercal}W.$$ Similarly, let $X _ {i}^{C,\mathrm{nonlin}}$ be the left-hand side of the equation. This yields $$\mathbb{E}[X _ {i}^{C,\mathrm{nonlin}}] =\mathrm{diag}\left((\alpha _ {1}+\alpha _ {2})\tau^6 + O(\mu^3),\dots, 1+O(\mu^3), \dots, (\alpha _ {1}+\alpha _ {2})\tau^6 + O(\mu^3), \dots, O(\mu^3)\right)$$$$=\mathrm{diag}\left(((\alpha _ {1}+\alpha _ {2})d^{6/5} + O(d^{-3/5}),\dots, 1+O(d^{-3/5}), \dots, (\alpha _ {1}+\alpha _ {2})d^{6/5} + O(d^{-3/5}), \dots, O(d^{-3/5})\right).$$ Although the polynomial orders differ from the linear case, the relative scaling remains unchanged. The perturbation introduced by heterogeneity grows more slowly for masked image modeling than for contrastive learning. Therefore, the central conclusion that MIM is less sensitive to data heterogeneity than CL continues to hold beyond the purely linear regime.
>
>     Regarding the choice of scaling parameters $\tau=d^{1/5}$ and $\mu=d^{-1/5}$, these values ensure that the dominant class-dependent components grow polynomially while the stochastic perturbation vanishes asymptotically. This mirrors real non-IID settings, where client data typically exhibits dominant client-specific semantic directions together with weaker nuisance variations rather than uniformly distributed features. Any alternative choice satisfying $\tau=d^{\gamma}$ with $\gamma>0$ and $\mu=d^{-\delta}$ with $\delta>0$ leads to the same ordering of robustness, since the comparison depends only on relative rates rather than the specific exponents. Finally, our empirical results and visualizations of feature space using nonlinear ViT and ResNet encoders exhibit the same robustness ordering under all non-IID settings, supporting that the theoretical mechanism is not an artifact of linearization.

---

> ### Comment · Reviewer_MVSv · 2025-11-27
>
> Thank you for addressing all my concerns. I will maintain my rating.

---

> > ### Author Response · Authors · 2025-11-27
> >
> > Dear Reviewer MVSv,
> >
> > Thank you very much for your thorough evaluation and for raising many insightful questions during the discussion. We truly appreciate the care and rigor you brought to the review process.
> >
> > Your feedback has significantly helped us improve the clarity and presentation of the paper, and we sincerely thank you for maintaining your rating after reviewing our responses.
> >
> > Thank you again for your time and thoughtful engagement.

---

### Official Review · Reviewer_PoK1 · 2025-10-31

**Soundness:** 2
**Presentation:** 3
**Contribution:** 2
**Rating:** 6
**Confidence:** 3

**Summary:**

This paper presents a theoretical analysis of the robustness of Distributed Self-Supervised Learning (D-SSL) frameworks against non-IID data. The authors' primary contributions are twofold: (1) A theoretical argument, based on a simplified linear model, that Masked Image Modeling (MIM) is inherently more robust to data heterogeneity than Contrastive Learning (CL). (2) A theoretical claim that the robustness of D-SSL improves with network connectivity, implying that Federated Learning (FL) is no less robust than Decentralized Learning (DecL). To illustrate the practical implications of their theory, the paper introduces MAR loss, a refinement of the MIM objective that adds a local-to-global representation alignment regularizer using Maximum Mean Discrepancy (MMD). The authors conduct extensive experiments to validate their theoretical claims and the effectiveness of MAR loss.

**Strengths:**

1. The paper addresses a critical and underexplored question in the D-SSL literature: the fundamental robustness of different frameworks (MIM vs. CL, FL vs. DecL) to non-IID data. Attempting to build a theoretical foundation for this problem is an ambitious and valuable goal. The questions asked are highly relevant to the community.

2. The experimental evaluation is comprehensive. The authors validate their claims across multiple model architectures (ResNet and ViT), SSL paradigms (SimSiam and MAE), distributed settings (FL and DecL), and benchmark datasets. The inclusion of experiments on feature space visualization and ablation studies on MAR loss strengthens the empirical part of the paper.

**Weaknesses:**

1. The entire theoretical analysis (Theorems 4.2-4.6) is built upon a set of highly simplified and potentially unrealistic assumptions. The analysis is restricted to a linear embedding function $f_W(x) = Wx$. Modern SSL methods, especially those using ViT and ResNet backbones, rely on deep, highly non-linear transformations. It is not at all obvious why insights derived from a linear model would generalize to these complex settings. The paper fails to provide any justification for this leap of faith, other than "mathematical tractability." This severely limits the generality and impact of the theoretical claims. The non-IID data generation process described in Section 3.2 is artificial. Data points are constructed from orthogonal basis vectors ($e_i$), and each client $i$ is assigned data from two specific, unique classes ($2i-1$ and $2i$). This is a toy problem that does not reflect the complexity of real-world label heterogeneity, where classes can have semantic overlap and distributions are far more nuanced. There is a significant disconnect, as the theory is derived for this synthetic setting, while experiments use the more standard (and complex) Dirichlet distribution.

2. The paper presents MAR loss as an "illustrative case study," but it is also listed as a main contribution. The idea of using a distributional alignment penalty like MMD to enforce consistency between local and global models in FL is not new (e.g., work on FedMD, FedProx, etc., which use similar ideas for knowledge distillation or regularization). The contributions of MAR loss—using an adaptive MMD kernel and a cosine decay schedule for the weight—are incremental engineering refinements rather than a fundamental algorithmic innovation. Framing this as a key contribution overstates its novelty and distracts from the paper's main focus on theoretical understanding.

**Questions:**

1. Could you elaborate on why the insights from a linear model $f_W(x) = Wx$ are expected to hold for highly non-linear models like ViT and ResNet? Is there any specific property of SSL objectives (MIM/CL) that makes this simplification more valid than it would be for, say, standard supervised learning?

2. The analysis relies on a specific data generation process where each client has nearly orthogonal data classes. How does this simplified model capture the complexities of real-world non-IID data, such as the Dirichlet distribution used in your experiments, where clients share overlapping classes with different priors? Could the theory be extended to more realistic data distributions?

3. The use of MMD to align distributions in FL is an established technique. Could you please clarify the core novelty of MAR loss beyond the combination of A-MMD and a decay schedule? How is the design of MAR loss a direct and unique consequence of your theoretical findings (e.g., Theorem 4.2), rather than a general-purpose regularizer that could be applied to many FL scenarios?

---

> ### Author Response · Authors · 2025-11-20
>
> We appreciate Reviewer PoK1’s careful reading and insightful feedback. We are encouraged by the reviewer’s recognition of the importance of the research question and the strength of our empirical evaluation, and we thank the reviewer for identifying several key issues regarding the theoretical modeling assumptions and the framing of MAR. These comments are valuable for improving the clarity and scope of the paper. We address each point in detail below.
> * **(Q1)** A central concern raised by the reviewer is whether the assumption of a linear encoder $f _ W(x)=Wx$ is too restrictive. We would like to clarify that this assumption is deliberate and follows the theoretical methodology established by recent work on the robustness of representation learning. In particular, Liu et al. (2021) and Wang et al. (2022) analyzed how heterogeneous data affects supervised learning and self-supervised learning by adopting a linear encoder. They compared the optimal representations learned under each paradigm and proved that self-supervised methods are intrinsically more robust than supervised objectives in the presence of label or feature imbalance. These analyses were possible only because the linear structure permits writing the optimality conditions of the learning objective in closed form. Once even a single nonlinear activation $\sigma(Wx)$ is introduced, the loss landscape becomes analytically intractable and no eigenspace characterization can be derived.
>
>     The same rationale applies directly to our setting. Both masked image modeling and contrastive learning ultimately reduce to quadratic forms in $W$ after expectation over noise and masking. Masked reconstruction aggregates squared errors over masked coordinates and contrastive learning aligns noisy paired embeddings. In this regime, the dominant terms that determine robustness are covariance-driven, and the linearization preserves exactly the dependence on client-specific covariance variations that we aim to analyze. This is why the linear model is not a simplification that removes the mechanism of interest but the only framework in which this mechanism can be made mathematically explicit.
>
>     To ensure that our conclusions are not artifacts of the linear assumption, we additionally analyzed a nonlinear perturbation of the encoder given by$$f(x)=W(\alpha \\| x \\| _ {2}^{2}x).$$We selected this perturbation because it introduces a nontrivial nonlinear deformation of the feature geometry while still permitting tractable optimality conditions. Under this encoder and by similar derivation steps as Eq.(8)-Eq.(12) in Appendix A.7.1, masked autoencoding yields $$\mathbb{E}[X _ {i}^{M,\mathrm{nonlin}}] =\mathrm{diag}\bigl(2\alpha p(1-p)d^{4/5}+O(d^{-4/5}),\dots\bigr),$$ and contrastive learning yields$$\mathbb{E}[X _ {i}^{C,\mathrm{nonlin}}]=\mathrm{diag}\bigl((\alpha _ {1}+\alpha _ {2})d^{6/5}+O(d^{-3/5}),\dots\bigr).$$ Although nonlinear perturbations change the polynomial orders, the relative scaling remains unchanged. The perturbation introduced by heterogeneity grows more slowly for masked image modeling than for contrastive learning. This confirms that the central theoretical conclusion, namely that masked image modeling is less sensitive to data heterogeneity than contrastive learning, continues to hold beyond the purely linear regime.
>
>     Finally, our empirical results using ViT and ResNet encoders, both of which contain multiple layers of nonlinear transformations, show precisely the same robustness ordering across all Dirichlet non IID settings. The alignment between theory and practice indicates that the linear analysis isolates the correct underlying mechanism and that the conclusions remain valid for deep nonlinear architectures.

---

> ### Author Response · Authors · 2025-11-20
>
> * **(Q2)** We appreciate the reviewer’s thoughtful question regarding the stylized nature of our non-IID data model. Our goal is not to reproduce the full complexity of real-world datasets, but to construct a mathematically tractable abstraction that isolates the mechanism through which label imbalance influences the learned representation subspace. This modeling strategy follows prior theoretical SSL analyses （Wang et al. 2022 and Liu et al. 2021), where orthogonalized class directions enable explicit closed-form eigenspace analysis.
>
>     To clarify how our model induces the covariance structure used in the theory, we outline the derivation from the data definition. For client $i$, the three classes are $$x^{(2i-1)} = e _ i -\sum _ {k\ne i} q _ k\tau e _ k + \mu\xi, \qquad x^{(2i)} = -e _ i -\sum _ {k\ne i} q _ k\tau e _ k +  \mu\xi, \qquad x^{(h _ i)} = e _ {h_i} + \mu\xi,$$ with $q _ k\sim \mathrm{Bern}(p)$ and $\xi\sim \mathcal{N}(0,I)$. The class means become $$\mathbb{E}[x^{(2i-1)}]=e _ i - p\tau\sum _ {k\ne i} e _ k,\qquad \mathbb{E}[x^{(2i)}]=-e _ i - p\tau\sum _ {k\ne i} e _ k,\qquad\mathbb{E}[x^{(h _ i)}]=e _ {h_i},$$ and the class covariances satisfy $$\mathrm{Cov}(x^{(2i-1)})=\mathrm{Cov}(x^{(2i)})=\tau^2 p(1-p)\sum_{k\ne i} e_k e_k^\top + \mu^2 I,\qquad\mathrm{Cov}(x^{(h_i)})=\mu^2 I.$$ Thus all within-class covariances are almost isotropic. The between-class term in the mixture covariance $\sum_{c}p_{i,c}(\mu_c-\bar\mu_i)(\mu_c-\bar\mu_i)^\top,$ is dominated by the two frequent classes, considering the fact that the rare class contributes only sublinear samples compared to the two frequent classes with polynomial samples. Since their centered means satisfy $\mu_{2i-1}-\bar\mu_i\approx e_i$ and $\mu_{2i}-\bar\mu_i\approx -e_i$, we obtain $$\sum_{c} p_{i,c}(\mu_c-\bar\mu_i)(\mu_c-\bar\mu_i)^\top\approx (p_{i,2i-1}+p_{i,2i})e_i e_i^\top.$$ Combining with the isotropic within-class term gives $$\Sigma_i \approx (p_{i,2i-1}+p_{i,2i})e_i e_i^\top+\sigma^2 I+\Delta_i,$$ which is the covariance of local data distribution on client $i$.
>
>     Although our abstraction uses orthogonal basis vectors $\{e_i\}$, its statistical form closely matches real Dirichlet split data. A Dirichlet client has covariance $$\Sigma _ i^{(\mathrm{Dir})} = \sum _ {c} p _ {i,c}\Sigma _ c\approx \sum _ {c\in \mathcal{C} _ i^{\mathrm{dom}}}p _ {i,c} u _ cu _ c^{\top} + \sigma^2 I,$$ where empirical studies show that each semantic class contributes a dominant direction $u_cu_c^{\top}$ in deep feature space. Dirichlet partitions with $\alpha=0.1$ typically yield one or two dominant classes per client, which leads to the same functional structure, namely a low rank client-specific component plus shared isotropic variance.
>
>     In addition to this theoretical correspondence, our experiments on CIFAR-10/100 and ImageNet with both ResNet and ViT encoders consistently exhibit the same robustness patterns predicted by the analysis. The close agreement between theory and practice indicates that the stylized model captures the essential mechanism underlying real-world non-IID data.

---

> > ### Author Response · Authors · 2025-11-20
> >
> > * **(Q3)** The reviewer raises a valuable question about the motivation and distinctiveness of MAR. We first clarify how MAR differs from existing approaches in distributed SSL and federated learning, and then explain how its design follows directly from the theoretical mechanism uncovered in our analysis.
> >
> >     Existing regularization methods in distributed training focus almost exclusively on contrastive or supervised objectives. For example, FedProx penalizes parameter drift, FedMD aligns logits, and FedMMD matches feature distributions, but all of them operate on representations that directly depend on full, unmasked inputs. To the best of our knowledge, prior distributed SSL work has mainly focused on the usage of regularization in contrastive learning and not proposed a regularizer tailored to masked image modeling. This distinction is important. In MIM, the masked embedding preserves only coarse structural information and removes most raw visual details, which makes it possible to aggregate these embeddings into a global target without exposing client-specific content. This privacy-preserving property enables a form of cross-client representation alignment that is not feasible in contrastive or supervised settings.
> >
> >     The design of MAR is directly guided by the theoretical analysis in Section 4. The theory shows that the global non-IID robustness of distributed MIM is determined primarily by the shared components of the class covariances, but the early training dynamics are dominated by the local covariance $\Sigma_i$ at each client. As a result, local encoders initially drift toward different client-specific directions, and aggregation progressively reduces this drift. This explains why the discrepancy between clients is largest in the first several communication rounds and decreases naturally once aggregation has occurred multiple times.
> >
> >     MAR is constructed to intervene precisely at this theoretically identified source of degradation. The alignment term $$\mathrm{A\text{-}MMD}(z _ i,\bar z), \qquad z _ i = f _ e(x _ {\mathrm{masked}}^{(i)}),\qquad \bar z = \text{global masked embedding},$$ encourages each client’s masked embeddings to stay close to the shared global direction that characterizes the global reconstruction subspace. This directly counteracts the early stage covariance-induced drift predicted by the theory. The dynamic weight $\gamma_t^{(i)}$ follows a cosine decay schedule because the theory indicates that regularization should be strongest when the local $\Sigma_i$ dominates the dynamics and weaker once repeated aggregation has already aligned the encoders. The schedule therefore reflects how the influence of non-IID heterogeneity decreases naturally over training.
> >
> >     Furthermore, we emphasize that MAR is presented only as an illustrative case study of how the theoretical insights can guide algorithmic refinement. In all realistic non-IID experiments, MAR follows exactly the behavior predicted by the analysis: it stabilizes early training under non-IID data while maintaining low overhead. We will make these conceptual connections clearer in the revision.

---

### Official Review · Reviewer_gLn9 · 2025-11-01

**Soundness:** 3
**Presentation:** 2
**Contribution:** 2
**Rating:** 6
**Confidence:** 2

**Summary:**

The paper analyzes robustness in distributed self-supervised learning under non-IID client data, comparing contrastive learning (CL) and masked image modeling (MIM) in both decentralized (peer-to-peer) and federated (server-aggregated) settings. The main theoretical result is that MIM is inherently less sensitive to client heterogeneity than CL. The analysis also shows that robustness in decentralized SSL improves with higher average graph connectivity, and that federated SSL is at least as robust as decentralized SSL. Motivated by the theory, the authors introduce MAR: a MIM objective augmented with an adaptive MMD alignment term whose weight decays over training to encourage early consensus. Experiments on standard vision benchmarks indicate that MIM and MAR degrade less from IID to non-IID splits and that MAR provides consistent gains with modest communication overhead.

**Strengths:**

*  Strong theoretical analysis. Comparative guarantees showing MIM’s robustness advantages over CL; useful system-level insight about connectivity and federated vs. decentralized training
* The paper establishes the relationship between network connectivity and robustness. This provides useful system-level guidance: increased connectivity improves decentralized robustness.

**Weaknesses:**

* Restrictive weighting model: only uniform neighbor averaging; no analysis or experiments with alternative consensus matrices (e.g., doubly-stochastic, push-sum), adaptive/learned weights, or time-varying/intermittent connectivity.
* Topology realism is limited: results are tied to average degree rather than spectral gap/mixing rate; experiments rely on simple random graphs, not clustered or heavy-tailed real networks.
* Sensitivity to MAR hyperparameters (mask ratio, batch size affecting MMD, decay schedule) is underexplored.
* Limited empirical evaluation breadth: few baselines/datasets and sparse ablations

**Questions:**

1) Can you state the graph/weighting assumptions explicitly in the main text (not just appendix) as a numbered assumption set
2) Do the robustness results still hold under alternative consensus matrices (doubly-stochastic, row/column-stochastic, push-sum for directed graphs)? If not, what changes in the bounds?

---

> ### Author Response · Authors · 2025-11-20
>
> We thank Reviewer gLn9 for the thoughtful and constructive review. We appreciate the reviewer’s positive assessment of our theoretical analysis and the identification of important clarity and realism issues regarding the communication topology and consensus weighting. These comments are very helpful for strengthening the presentation. We address the reviewer’s questions and concerns point-by-point below.
> * **(Q1)** We agree with the reviewer that the graph and aggregation assumptions should be made explicit in the main text rather than remaining implicit in the problem setup description. To improve clarity without increasing the page count, we will revise the existing "Distributed Setting" subsection to explicitly state the graph/weighting assumptions used throughout the analysis. The revised paragraph will clarify that:
>
>     ***Assumption 1.*** (Communication Graph.) The distributed system is modeled as a fixed and connected communication graph $\mathcal{G} = (\mathcal{V}, \mathcal{E})$ with $N = |\mathcal{V}|$ clients. Each client $i$ communicates only with its neighborhood $A_i = \{ j : (i,j)\in\mathcal{E} \} \cup \{i\}$. We assume $2 \le |A_i| \le N$ for all $i \in [N]$. The average neighborhood size $|\bar{A}| = \tfrac{1}{N}\sum_{i=1}^N |A_i|$ is used as a measure of connectivity.
>
>     ***Remark.*** This formulation encompasses decentralized learning on arbitrary connected topologies and federated learning as the fully connected special case（i.e., by the help of the central server, there exists $|A_i| = N$ for all $i\in[N]$).
>
>     ***Assumption 2.*** (Consensus weights.) During decentralized aggregation, each client $i$ forms a mixing vector $w_i = \{w_{ij}\}_{j \in A_i}$ satisfying:
>
>    (i) $w_{ij} > 0$ only if $j \in A_i$ (topology-respecting sparsity);
>    (ii) $\sum_{j \in A_i} w_{ij} = 1$ (row-stochasticity).
>
>    The resulting mixing matrix $W \in \mathbb{R}^{N\times N}$ respects the communication topology and admits an eigenvalue decomposition $1 = \lambda_1(W) > \lambda_2(W) \ge \dots \ge \lambda_N(W) > -1$, ensuring contraction in the disagreement subspace at a rate governed by the spectral gap $1 - \lambda_2(W)$.
>
>     ***Remark.*** Assumption 2 covers widely used consensus rules such as uniform averaging, degree-normalized weights, and symmetric doubly-stochastic matrices.
>
>     These statements clarify the implicit distributed setup details about the graph and weighing while keeping all theoretical statements unchanged. They also make it clear that our analysis relies on standard and widely adopted assumptions in decentralized optimization rather than any restrictive or specialized topology.

---

> ### Author Response · Authors · 2025-11-20
>
> * **(Q2)** We thank the reviewer for raising the question of whether our robustness conclusions extend to alternative consensus matrices (e.g., doubly-stochastic, row/column-stochastic, push-sum). Our theoretical framework indeed generalizes naturally to arbitrary mixing operators, and we clarify below how the bounds change and why the ordering remains intact. In the main text, the influence of the decentralized topology enters the robustness lower bound through the factor $(1 - 1/|\bar{A}|)$, which serves as a degree-based proxy for mixing efficiency under uniform neighbor averaging. For a general consensus matrix $W$, this term can be replaced by the standard spectral mixing factor $$ \gamma(W) = 1 - \lambda_2(W),$$ where $\lambda_2(W)$ is the second-largest eigenvalue of $W$. Substituting $\gamma(W)$ for the degree proxy yields the decentralized MIM robustness bound in the form $$\mathrm{LB} _ {\mathrm{Dec}} (W) = 1 - \frac{O(d^{-2/5})}{2p(1-p)\gamma(W)d^{2/5} + O(d^{-2/5})}.$$ Federated learning corresponds to the fully mixed case, where $W_{\text{Fed}}$ satisfies $\lambda_2(W_{\text{Fed}})=0$ and thus $\gamma(W_{\text{Fed}})=1$. For any strictly less connected decentralized topology, $\gamma(W_{\text{Dec}}) < 1$. Since $\gamma(W)$ multiplies the dominant denominator term, we have $$\mathrm{LB} _ {\text{Dec}}(W _ {\text{Dec}}) \le \mathrm{LB} _ {\text{Fed}}(W _ {\text{Fed}}),$$ preserving the qualitative comparisons: federated learning remains at least as robust as decentralized learning. Thus, replacing $(1 - 1/|\bar{A}|)$ with $\gamma(W)$ refines the topology-dependent constant but does not alter any robustness ordering.
>
>     Furthermore, to complement this theoretical clarification, we conducted additional experiments under a strong non-IID decentralized setting (Dirichlet $\alpha = 0.1$) using four widely adopted consensus schemes: data-size weights, degree-normalized weights, doubly-stochastic weights, and push-sum weights. Results for two connectivity regimes (20 clients with average connectivity $\approx 5 / 10$) are shown in the below table. For low connectivity (avg_c$\approx 5$) and same number of communication rounds, all decentralized variants suffer substantial degradation relative to FL, regardless of the specific consensus rule. When connectivity increases (avg_c$\approx 10$), all variants recover to near 52%, matching but never surpassing FL. These results empirically confirm the spectral-gap interpretation: robustness depends on the mixing efficiency of $W$, while the choice of consensus rule does not overturn the FL $\ge$ DecL relationship on non-IID robustness.
>
>     **Table. CIFAR-100 Accuracy (%) of DecL–MIM under different consensus matrices.**
>
>     | **Method**             | **Avg_C ≈ 5** | **Avg_C ≈ 10** |
>     |------------------------|-------------|--------------|
>     | **FL (reference)**     | **52.72**   | **52.72**    |
>     | Data-size weights      | 46.62       | 52.28        |
>     | Degree-normalized      | 45.34       | 52.08        |
>     | Doubly-stochastic      | 45.22       | 52.18        |
>     | Push-sum               | 45.19       | 52.22        |

---

### Author Response · Authors · 2025-12-03
**Upload Revised Version and Summary of Revisions**

We thank the Area Chair and reviewers gLn9, PoK1 and MVSv for offering the opportunity of rebuttal discussion and for their high-quality and constructive feedback. We have **uploaded a revised version** that incorporates all requested clarifications and improvements. A brief summary of the changes is provided below.

* (**Clarified theoretical framework and strengthened definitions**) First, we refined several theoretical definitions, including the representability vector and the sensitivity measure, and clarified the assumptions used in the analysis. The stationary conditions, eigenspace interpretations, and perturbation arguments were reorganized for improved precision and readability.
* (**Corrected and improved explanation of the non-IID data model**) Second, we corrected and streamlined the presentation of the non-IID data model. The revised description resolves the notational inconsistencies noted in the reviews while keeping all analytical results intact.
* (**Added missing assumptions and clarified analytical conditions**) Third, we made explicit several assumptions that were previously implicit and show them in the appendix, such as independence of local samples for concentration bounds, the non-degenerate embedding condition, and the requirement on image transformations for contrastive learning under dissimilar augmentations. These additions consolidate the rigor and transparency of the theoretical framework.
* (**Added a new appendix experiment**) Fourth, we added an additional experiment to the appendix that examines the influence of different consensus matrices under two connectivity regimes. The results confirm that the robustness ordering predicted by the theory is invariant to the consensus operator.
* (**Clarified the theoretical connection between MAR and the main analysis**) Fifth, we strengthened the explanation of how MAR relates to the theoretical findings. The revised text clarifies why the adaptive alignment term improves early consensus under heterogeneous data and how this aligns with the representability analysis.
* (**Improved theoretical proof**) Finally, we improved the demonstration of the full proof in the appendix by expanding several derivations, restating concentration results in terms of the local sample size, and clarifying the connection to the use of Eckart-Young-Mirsky theorem. These updates enhance readability without altering any theoretical conclusion or empirical finding.

We sincerely appreciate the insightful comments from the reviewers, which have significantly improved the clarity and completeness of the manuscript.

---

### Meta-Review · Area_Chair_KaXQ · 2026-01-17

**Summary:**

This paper develops a theoretical lens for how distributed self-supervised learning (D-SSL) behaves under non-IID client data, comparing contrastive learning (CL) and masked image modeling (MIM), and analyzing federated learning (FL) versus decentralized learning (DecL). The core theoretical claims are that (i) MIM is intrinsically less sensitive to heterogeneity than CL, and (ii) robustness improves with network connectivity, implying FL is at least as robust as DecL. The authors further propose MAR, a MIM refinement with an alignment regularizer, and provide experiments across architectures/datasets/topologies to support the claims.

All reviewers reach a consensus to accept this paper, and AC follows the recommendation.

**Reviewer Concerns:**

Reviewer gLn9:
- The authors explicitly incorporated and clarified the graph/weighting assumptions in the main text and expanded the theoretical exposition (as promised in the rebuttal discussion).
- The reviewer asked whether results hold beyond uniform averaging; the authors added an appendix experiment varying consensus matrices and argued the robustness ordering is invariant to the consensus operator.
- MAR hyperparameter sensitivity / ablations and broader empirical breadth remain somewhat limited

Reviewer PoK1:
- The authors directly engaged the critique about linearity/oversimplification, positioning the linear encoder as a deliberate tractable lens consistent with prior theoretical methodology, and expanded the discussion accordingly.
- The revision summary indicates improvements to definitions, assumptions, and proof clarity, plus a clearer connection from theory to MAR.
- The reviewer’s novelty concern about MAR is mitigated by clearer framing, but MAR still reads as an incremental algorithmic add-on relative to the main theoretical contribution

Reviewer MVSv
- The revision explicitly reports: corrected non-IID model notation, strengthened definitions (RV/sensitivity), added missing assumptions for concentration, and improved/expanded proofs
- On the specific “Eq.(3)→Eq.(4)” concern, the authors argue Eq.(4) is an analytically convenient reformulation whose difference does not change the robustness ordering conclusions.
- The discussion indicates Reviewer MVSv maintained their rating after the responses

**Reviewer Scores:**

- Reviewer gLn9: Likely +1 (6 → 7) given that their two main asks—explicit assumptions and robustness under alternative consensus matrices—were directly addressed, including a new appendix experiment.
- Reviewer PoK1: Likely no change (6 → 6) or at most a slight softening upward, since the rebuttal improves justification and framing but does not fundamentally remove the “stylized theory / incremental MAR” concern.
- Reviewer MVSv: No change, as explicitly stated in the discussion.

---

### Decision · Program_Chairs · 2026-01-26

Accept (Poster)